cognition/behaviour

learning, cognitive ecology, individual differences, great tit, radio frequency identification

**Authors for correspondence:**
Michael S. Reichert
e-mail: michael.reichert@okstate.edu
John L. Quinn
e-mail: j.quinn@ucc.ie

# Multiple factors affect discrimination learning performance, but not between-individual variation, in wild mixed-species flocks of birds

Michael S. Reichert[1,2], Sam J. Crofts[3], Gabrielle L. Davidson[2,5], Josh A. Firth[3,4], Ipek G. Kulahci[2] and John L. Quinn[2]

[1]Department of Integrative Biology, Oklahoma State University, Stillwater, OK, USA
[2]School of Biological Earth and Environmental Sciences, University College Cork, Cork, Ireland
[3]Edward Grey Institute, Department of Zoology, and [4]Merton College, University of Oxford, Oxford, UK
[5]Department of Psychology, University of Cambridge, Cambridge, UK

MSR, 0000-0002-0159-4387; GLD, 0000-0001-5663-2662; JAF, 0000-0001-7183-4115; IGK, 0000-0003-0104-0365

Cognition arguably drives most behaviours in animals, but whether and why individuals in the wild vary consistently in their cognitive performance is scarcely known, especially under mixed-species scenarios. One reason for this is that quantifying the relative importance of individual, contextual, ecological and social factors remains a major challenge. We examined how many of these factors, and sources of bias, affected participation and performance, in an initial discrimination learning experiment and two reversal learning experiments during self-administered trials in a population of great tits and blue tits. Individuals were randomly allocated to different rewarding feeders within an array. Participation was high and only weakly affected by age and species. In the initial learning experiment, great tits learned faster than blue tits. Great tits also showed greater consistency in performance across two reversal learning experiments. Individuals assigned to the feeders on the edge of the array learned faster. More errors were made on feeders neighbouring the rewarded feeder and on feeders that had been rewarded in the previous experiment. Our estimates of learning consistency were unaffected by multiple factors, suggesting that, even though there was some

influence of these factors on performance, we obtained a robust measure of discrimination learning in the wild.

# 1. Introduction

Animal cognition provides a framework for understanding evolutionary processes operating on functional behavioural variation [1,2]. Demonstrating consistent between-individual differences in cognitive performance is fundamental, as such evidence would point to long-term implications for individual behaviour because, for example, this suggests the trait may be heritable and has the potential to evolve in response to natural selection [3]. However, quantifying individual differences in cognitive performance remains challenging (reviewed by the authors in [4–7]). One reason for this is that, while many studies aim to measure cognitive ability, which is the inherent effectiveness of an individual's cognitive mechanism, cognitive ability cannot be directly measured and is instead inferred from an individual's performance on a cognitive task, which is subject to influence from myriad additional factors [5,8–10]. A related challenge is that the consistency, or repeatability, of cognitive traits can be difficult to estimate (reviewed by Cauchoix *et al.* [11]), both because repeatability estimates can be obscured by confounding variables [12] and because cognitive performance is influenced by previous experience and therefore cannot always be measured multiple times under standardized conditions [13]. An important step towards addressing these challenges involves quantifying individual performance using clearly defined cognitive assays, characterizing the underlying drivers and potential confounds, and doing so with the large sample sizes that are necessary for evolutionary and ecological investigations. Here, we studied individual performance in an initial discrimination learning experiment and two subsequent reversal learning experiments, and controlled for a range of cryptic confounding effects.

Multiple factors influence cognitive performance. For instance, individual attributes such as age, sex and species are related to both cognitive ability and motivation and ability to interact with the task [14–16]. However, the strength and direction of these effects is not consistent between studies [14–17], which suggests that other contextual variables often drive individual differences in cognitive performance. Cognitive performance is also affected by participation rates, which can introduce a major source of bias in studies conducted in wild populations. Few studies report participation rates, and when they do, they often report very low participation (reviewed by van Horik *et al.* [18]). For example, less competitive individuals may be less likely to participate, not because of differences in cognitive ability, but because they are excluded by others from accessing devices offering a reward [19,20]. Alternatively, less competitive individuals may be unable to exploit their primary food source and thus rely on their cognitive abilities to access resources from experimental devices (the 'necessity drives innovation' hypothesis; [21]).

Additionally, cognitive performance can be influenced by factors related to the characteristics of the experimental design and its interaction with the environment. Some of these factors may lead to biases (i.e. a non-random distribution of errors) in how individuals explore experimental arrays. For instance, distance to cover or location of the experimental apparatus relative to other profitable food sources [22] will influence individuals' participation and performance in experiments. Furthermore, in discrimination learning experiments in which multiple devices are placed in an array, some of which are rewarding and some of which are not, the placement of devices with respect to one another is likely to affect an individual's performance. Having to visit rewarding devices on the edge of an array may allow for easier discrimination because they are more distinctive having a neighbouring feeder only on one side. Edge locations may also be subject to greater predation risk [23], which can affect visitation rates and subsequent cognitive performance [22]. Previous experience is an important factor in reversal learning experiments, because having been rewarded in one location may make individuals more likely to continue to explore that location even after the contingencies have switched. Finally, social interactions may influence cognitive performance through a range of effects [24–27], such that experimental designs should account for or manipulate the social environment to gain a better understanding of individual variation in cognitive performance.

We investigated individual variation in learning performance in the wild in mixed-species flocks of two songbird species: great tits *Parus major* and blue tits *Cyanistes caeruleus*. The study species are well suited for studying cognition in the wild. They gather in large numbers during the winter when foraging [28] and are highly innovative foragers [24,29–31] that readily interact with cognitive tasks [32–35]. Discrimination learning is likely to be important for these species, because the ability to discriminate among rewarding

and non-rewarding food patches is not only central to making optimal decisions among animals in general [36], but is likely to be especially important during the harsh winter conditions that these species experience. Recent technological advances have opened possibilities to perform discrimination learning experiments in wild populations using automated feeder arrays [37–39]. We used an array of automated feeders to study individual variation in performance in an initial discrimination learning experiment, followed by two successive reversals in which each individual was assigned to a new rewarding feeder to examine individual flexibility and consistency in learning. Each experiment required individuals to learn to visit one feeder that would provide them with a reward in an array with four other unrewarding feeders (probably, but not necessarily, based on learning the rewarding feeder's location in the array).

We asked two main questions: first, which factors influenced individual performance in each learning experiment? Within this question, we first quantified participation rates for each experiment and examined the factors that influenced the likelihood of participation in the experiment. We then examined both individual (age, sex, species) and local environmental (location of the feeder within the array) effects on learning speed. Our aim in these analyses was not to test specific hypotheses about whether one type of individual would perform better than another, but rather to assess the influence of these putative confounds on our measures of cognitive performance. Finally, we examined whether there were biases in the patterns of errors made by individuals, to gain insights into how the array set-up, individual characteristics and previous experience influenced individual performance.

Second, was individual performance consistent across the three experiments, or at least across the two reversal experiments? The consistency of individual performance across these tasks gives insights into the role of individual variation in driving behaviours in nature, where reward contingencies change frequently. Discrimination ability may influence performance in both initial discrimination learning and reversal learning, but reversal learning also involves additional cognitive abilities that are not involved in learning the initial discrimination [40,41]. A negative correlation between initial discrimination performance and performance in a reversal experiment would suggest trade-offs between speed of learning and the flexibility necessary to adjust to new reward contingencies [42–45]. Alternatively, a positive correlation would suggest a single underlying trait controlling substantial variation in performance in both tasks [2]. However, estimates of consistency are potentially confounded by a wide variety of variables that could inflate or underestimate consistency [12,46]. Therefore, we also examine how our estimates of consistency were affected by whether or not we accounted for potential confounding factors.

With these analyses, we provide new insights into individual-level differences in cognitive abilities in the wild, and the confounding effects that should be controlled in the process. We use these insights to highlight how knowledge of individual differences is crucial for further understanding the evolutionary and ecological causes and consequences of cognitive abilities.

# 2. Methods

## 2.1. Study site and species

The study took place in Wytham Woods (51°46′ N, 1°20′ W), Oxfordshire, UK. Birds were captured with mist nets to be ringed, aged and sexed using a standard protocol that has been in place since the 1960s [47]. Great tits and blue tits were additionally fitted with plastic rings containing a passive integrative transponder (PIT) tag (IB Technology, Aylesbury, UK), allowing for individual identification of birds that interacted with our experimental devices (see below). We focused ringing efforts in winter 2017–2018 at locations where we would later place the experimental feeders. In addition to winter mist netting, many birds were previously marked because all breeding adults and nestlings are routinely captured and tagged during the spring breeding season [47].

## 2.2. Feeder array

We set out sunflower-seed feeder arrays in multiple locations chosen to maximize bird numbers and to minimize movement of individuals between arrays. We ran trials in four locations in November–December 2017, and in a different set of four locations in January–February 2018. Feeders were active each day during daylight hours from 07.00 to 16.30. Feeders shut down and visits were not recorded outside of these hours to save battery life. While some individuals may have continued to attempt to visit in any remaining daylight outside of these hours, possibly affecting their experience with the reward contingencies, we consider this unlikely to have substantially affected our results because there

was a large drop in visits towards the time that the feeders were closed, suggesting that few birds were attempting to feed at these hours (e.g. zero visits took place between 07.00 and 07.10 while 1629 visits took place between 08.00 and 08.10 in the initial learning experiment. Two feeders were inadvertently programmed to be active until 17.30, and in the initial learning experiment, these recorded only 94 total visits from 16.30 to 17.30, compared with 641 visits to the same two feeders from 15.30 to 16.30). Each feeder was equipped with a radio frequency identification (RFID) antenna placed near the single opening to the feeder, which served as a perch and recorded the individual ID and time of visit for each bird coming to the feeder. During the learning experiments, selective feeders were used [35,48], where access to food was controlled by a solenoid placed behind a transparent plastic door at the feeder opening. The solenoid would release upon detection of specific PIT tags, allowing only those birds assigned to a given feeder to obtain food by pushing open the door, which otherwise was immobile (see below). RFID readings and solenoid activation were controlled by a custom program loaded onto a printed circuit board ('Darwin Board', Stickman Technologies Inc., UK). All visits were monitored with the RFID antenna throughout the experiment. Each feeder was surrounded by a cylindrical cage of chicken wire, which had holes that were large enough for our study species to pass through but deterred squirrels and larger birds from accessing and damaging the feeder.

A depiction of the experimental timeline is given in electronic supplementary material, figure S1. Before the learning experiment started, we placed two feeders 50 m apart with feeder doors open and therefore accessible to all birds (5 days). The purpose of this portion of the experiment was to attract birds to the area and to begin habituating them to the feeders. After 5 days, we closed the feeder door so that feeders were only accessible to birds with PIT tags (7 days in 2017; 8 days in 2018; difference between years came because one 2018 site had low visit numbers at first and more time was needed to attract good numbers to the area). The purpose of this portion of the experiment was to continue to attract birds to the area while also allowing tagged birds to habituate to pushing the door to access food (at this point, all feeders were accessible to all tagged birds) and overcome any initial neophobia towards the devices. The door was transparent, so birds could see the food inside and, for individuals that were allowed access, pushing back the door and accessing the food reward occurred naturally as a result of their attempts to peck at the seeds. There was no problem-solving required to access the food reward. The next step was to remove the feeders at 50 m distance and create the array that was to be used during the learning trials: five feeders arranged in a straight line, each 1 m apart. This array was located at the midpoint between the original two feeders. Feeders were aligned parallel to natural cover at each study site, as distance from cover affects participation [22]. The layout of available cover necessitated a linear feeder array (rather than arranging feeders in a circle, which in preliminary trials caused severe biases in visits favouring those feeders closer to cover; M.S.R., I.G.K. & J.L.Q. 2017, unpublished data). We left feeders in this arrangement for 4 days, with all feeders again accessible to any PIT-tagged bird, with the aim to habituate birds to the new feeder arrangement. After this 4-day period, we began the learning experiments.

## 2.3. Learning and reversal learning

To measure individual variation in learning ability, we restricted each individual's access to only one of the five feeders in the array, by programming the particle control board with a list of PIT-tagged birds. Therefore, each feeder only opened the solenoid and allowed access to the food reward for specified birds, but the feeders recorded every visit of PIT-tagged birds, whether or not the bird was allowed to access the reward. Lists of individuals assigned to each feeder were generated from a database of birds that had visited the site during the pre-learning trials or had been captured in the area at any point. We randomly assigned birds from each species separately to ensure a similar distribution of species across the feeders. Within each species, individuals were randomly assigned to feeders to minimize interactions between individuals that might have been socially connected.

The initial learning phase (hereafter, 'initial learning') ran for 8 days, followed by two reversal learning phases (hereafter 'first reversal' and 'second reversal') in which each bird was assigned to a new feeder in each phase. We used one of two treatments per site to determine how individuals were reassigned to new feeders during reversal learning, with the aim to test for potential social influences acquired during initial learning that may influence reversal learning speed and to manipulate the potential for social learning. For the purposes of the present manuscript, this social group treatment is mentioned and included only to control for the effect. More extensive analysis of this effect is beyond the scope of our aims here and forms part of a separate paper in preparation. In the 'stable' social group treatment (four randomly selected sites), the entire cohort of birds assigned to a given feeder

during initial learning was reassigned to the same new feeder. In the 'unstable' social group treatment (the other four sites), cohorts assigned to a given feeder during initial learning were broken up, and each individual bird was reassigned to a new feeder. The second reversal followed the stable and unstable treatments as described in the first reversal. In all cases, birds were reassigned to a different feeder from the one they were assigned to in the initial learning. Each reversal experiment was performed for 8 days in 2017 (two sites for each social group treatment) and, owing to operational differences, 10 days in 2018 (two sites for each social group treatment).

We note that our design does not exactly replicate the classical tests of reversal learning where contingencies are switched between only two options (e.g. [49–51]), but the underlying principle of our experiment is similar in that reward contingencies change: subjects must stop going to a previously rewarded feeder and switch to a new rewarded feeder. Reversal learning paradigms sometimes do encompass more than two possible choices [52]. Reversal took place on the same day for all birds because of the social stability treatment and because we could only extract learning performance after the data were downloaded from the devices, which stored data independently of one another. Individuals that had not learned during the initial learning were removed from the analyses related to learning speeds (see table 1 for sample sizes).

## 2.4. Data analysis

We performed five main analyses on the experimental data: first, we investigated what determined whether an individual participated in the learning experiments. Second, we evaluated the robustness of our chosen learning criterion by comparing it with several alternatives. Third, for those birds that participated, we investigated what factors determined the speed of learning and reversal learning. Fourth, we analysed individual consistency in performance (learning speed) across the different experiments. Fifth, we examined in detail the spatial pattern of errors made during the experiments to gain insights into the processes involved in exploration of the array and potential biases.

## 2.5. Visits and data inclusion

The raw dataset consisted of rows containing the date, time and PIT tag for each detected visit at each feeder. We considered consecutive detections of the same bird to the same feeder within 2 s of each other to be a single visit [53]. Our analyses were restricted to great tits and blue tits; a range of other species were recorded in very small numbers and excluded from further analyses. Because many of our analyses examined effects of sex, we excluded individuals for whom sex was unknown (because the individual had not been recaptured since being ringed as a nestling; $N = 18/269$ blue tits and $11/169$ great tits were of unknown sex).

Some individuals were not assigned a feeder because they had not been previously detected at the site until the learning experiment began; these individuals were excluded from all analyses. A total of 21 individuals that were included in the analyses appeared at more than one site (including individuals occurring at more than one site over the 2017 and 2018 repeats). In almost all cases, these individuals appeared frequently at one site and very rarely at any others. Therefore, to avoid pseudoreplication, for these birds, we only analysed data from the single site it was recorded most frequently at. When birds participated in both 2017 and 2018, we simply excluded them from the 2018 data and chose the site visited in 2017 (because this was the individual's first encounter with the feeders). Unless otherwise noted, all analyses were performed separately for the three different learning experiments.

## 2.6. Participation

A bird was considered to be a participant in an experiment if it visited the feeders at least 50 times during that experiment. One bird in the initial learning experiment reached the learning criterion (see below) but visited less than 50 times and this individual was also classified as a participant. All other birds that visited less than 50 times were classified as non-participants. We chose 50 as the cut-off for participation because only one of 21 birds that visited between 20 and 49 times met the initial learning criterion, while five of nine birds that visited between 50 and 79 times met the initial learning criterion. Individuals were only considered participants in the subsequent reversal learning experiment if they had met the participation and learning criteria in the previous experiment, and then again visited 50 or more times during the reversal experiment.

**Table 1.** Sample sizes in each component of the experiments. Each row gives the number of individuals of the species/sex/age class that were included in the different analyses for each of the three experiments. Note that each column is a nested subset of the individuals in the previous column, and this nesting applies across all three experiments. For instance, for the appearances in the first reversal learning, only those birds that actually met the learning criterion in the initial learning experiment are included. If a bird failed to participate or learn in the previous experiment, it is not counted as appearing in future experiments even if it was actually present because such individuals were not included in analyses.

| | initial learning | | | first reversal learning | | | second reversal learning | | |
|---|---|---|---|---|---|---|---|---|---|
| | appearing | participating | learning | appearing | participating | learning | appearing | participating | learning |
| blue tit female adult | 52 | 25 | 24 | 24 | 23 | 22 | 22 | 22 | 22 |
| blue tit female juvenile | 45 | 33 | 31 | 30 | 28 | 27 | 24 | 24 | 24 |
| blue tit male adult | 86 | 40 | 38 | 38 | 37 | 36 | 33 | 32 | 32 |
| blue tit male juvenile | 68 | 41 | 38 | 38 | 38 | 38 | 37 | 37 | 37 |
| great tit female adult | 48 | 31 | 28 | 28 | 25 | 25 | 25 | 24 | 24 |
| great tit female juvenile | 24 | 15 | 12 | 11 | 10 | 9 | 9 | 9 | 9 |
| great tit male adult | 58 | 38 | 35 | 35 | 30 | 29 | 26 | 24 | 24 |
| great tit male juvenile | 28 | 17 | 15 | 14 | 13 | 12 | 11 | 11 | 11 |
| total | 409 | 240 | 221 | 218 | 204 | 198 | 187 | 183 | 183 |

To analyse the factors that determined participation in the initial learning experiment, we ran generalized linear mixed models fitted to a binomial distribution with a log link function, with the response variable of whether or not the individual met the criterion for participating, and included the following fixed effects: age (individuals were categorized as either adults or juveniles [less than 1 year old] because exact age was not known for all birds), sex, species, feeder location (edge, i.e. either end of the feeder array; or centrally located, the middle three feeders), and all two- and three-way interactions between age, sex and species, with site as a random term. We combined the middle three feeders into a single 'centre' category because our preliminary analyses indicated some differences in behaviour between the two edge feeders and the three centre feeders, but no differences in behaviour to the very central feeder compared with its two neighbouring feeders. We do not include similar analyses for the reversal learning experiments, because almost all individuals that participated and learned in the initial learning experiment continued to participate in subsequent experiments (table 1), and there was therefore an insufficient sample of non-participants for these experiments. We used the lme4 package [54] in R v. 3.5.2 software [55] to fit an initial model and performed a backwards stepwise procedure to remove non-significant terms beginning with the highest-order interactions and arrive at a final model. Model estimates for non-significant terms were calculated by adding them as a single additional variable to the final model.

## 2.7. Learning criterion

To determine when birds had learned the task, we assumed that each visit was an attempt to feed. Therefore, each visit a bird made to an unassigned feeder was counted as an error, and each visit to a feeder the bird was assigned to was counted as a correct choice. In order for a visit to be detected by the antenna and registered in our dataset, birds had to pass through the small holes of the cage surrounding each feeder and then land on a $5 \times 5$ cm antenna platform in front of the only access point to the feeder. Thus, birds were unlikely to land on this platform incidentally. We considered birds to have learned the task once they met the criterion of visiting the correct feeder 80% of the time on 20 consecutive visits (with the requirement that the first visit in that window be a correct visit) [56–58]. We evaluated the robustness of this criterion by comparing it with alternative criteria with different success percentages (80 or 90% correct) and numbers of consecutive visits (10, 20 or 30). We calculated learning speeds for each individual using each of these methods. The learning speed was the number of visits until the first visit at which the bird met the criterion defined above. We then used a correlation analysis to determine how similar the calculated learning speeds were under the six different possible criteria. High correlation coefficients would indicate that any of these criteria provide a similar answer and the measurement of learning speed is robust to the specific details of the chosen criterion.

To further evaluate our choice of learning criterion, we analysed the performance of individuals for all of their visits after they had reached the criterion. If our criterion captures the moment that the birds learned the task, then we would expect individuals' performance to remain high throughout the rest of the experiment. We therefore calculated the proportion of birds that maintained performance above 80% after having reached the learning criterion.

## 2.8. Learning speeds

We analysed what factors influenced learning speed, that is, the number of visits to meet criterion, for each of the three experiments. Individual learning speed was ln-transformed to meet assumptions of normality and analysed using linear mixed models with a Gaussian distribution. We included age, sex, species, the two- and three-way interactions between age, sex and species, feeder position and the average time interval between successive visits prior to meeting criterion (excluding time periods in which feeders malfunctioned, see below). The inter-visit interval was included to account for the fact that the amount of time elapsed between successive experiences often affects the speed of learning [59,60]. We could not directly control inter-trial interval as in many laboratory studies because birds' visits were voluntary and we did not restrict their rewards in any way. For the reversal experiments, we included social group treatment, learning speed in the previous experiment and the number of correct visits made after the learning criterion was met as additional fixed factors. As above, for reversal learning, we only included in analyses those birds that had participated and learned in the previous experiment. Some feeders malfunctioned during the course of the experiment and did not open for any of the birds or record any visits until they were repaired. Malfunctions occurred either because of failure of the antenna to register any visits or unexpected loss of power to the devices. We therefore included the duration of feeder

malfunction before the bird reached learning criterion for both the assigned (own) feeder and separately for any of the other feeders in that site as additional fixed effects. Effect estimates were calculated using restricted maximum likelihood.

## 2.9. Pattern of errors

We quantified the likelihood of different types of errors made by individuals that met the learning criterion to gain insights into how they were exploring the feeders and learning the task. For these analyses, as we were interested in how individuals' errors were distributed across the feeders, we excluded all correct visits (2 of 221 individuals that met the learning criterion in the initial learning experiment were therefore excluded because they never made an error). First, we asked whether there was an overall bias towards feeders on the edge or in the centre. We determined the null probability of an individual making an error at an edge feeder, under the assumption that errors were distributed randomly across the feeders. This probability is 50% (two incorrect edge feeders/four total incorrect feeders) for birds assigned to a feeder in the centre, and 25% for birds assigned to a feeder on the edge (one incorrect edge feeder/four total incorrect feeders). We then tested whether birds were biased towards the edge more than expected by subtracting the actual percentage of errors made at feeders on the edge from the null expectation for each bird. We used a one-way Wilcoxon test to determine if bias values differed significantly from zero. We then asked whether birds assigned to the edge were more biased towards errors on edge feeders than were birds assigned to the centre. This would be expected if birds on the edge were primarily learning by discriminating between the two edge feeders (i.e. learning to go to the left or to the right). We used a two-way Wilcoxon–Mann–Whitney (WMW) test to compare the bias towards the edge of birds that were assigned to the edge and birds that were assigned to the centre, and also to compare the bias between the two species.

We also asked whether errors were biased towards the feeders immediately adjacent to a bird's assigned feeder. We calculated the bias towards neighbouring feeders by subtracting the observed percentage of errors that were made at neighbouring feeders from the expected error rate based on the null probability of making an error at a neighbouring feeder if visits were distributed at random (which, as above, is 50% for birds assigned to a feeder in the centre, and 25% for birds assigned to a feeder on the edge). As above, we used a one-way Wilcoxon test to determine if bias values differed significantly from zero, and a two-way WMW to compare the bias towards neighbouring feeders of birds that were assigned to the edge or centre, and between great tits and blue tits.

Finally, for the reversal learning experiments, we tested whether individuals were biased to make errors at the feeder to which they were previously assigned in the prior experiment. In this case, the null expected percentage of errors at the previously assigned feeder was 25% for all birds. We used a two-way WMW to compare the bias towards previous feeders for great tits and blue tits.

## 2.10. Individual consistency in learning speed

Consistent differences between individuals in behaviour are typically quantified using repeatability coefficients generated from a mixed model analysis [46]. However, estimating repeatability in cognitive studies is challenging, because even when the same behavioural outcome is measured, the underlying cognitive processes may vary across trials [11]. In particular, initial learning and reversal learning probably involve different cognitive processes [40,41] and as a consequence may represent two separate traits. The advantage of the mixed model approach over related approaches, such as correlation analysis, is that it enables controlling for multiple confounding effects (both fixed and random) on both traits at the same time. We used mixed models and calculated repeatability coefficients to investigate whether there were consistent, between-individual differences in learning speed across the three experiments, but for the reasons expressed above and for heuristic reasons, we chose to refer to this as consistency in learning performance. We calculated individual consistency using the rptR package [12] in R, with ln-transformed learning speed as the response variable and individual ID as a random effect. We calculated consistencies that were both adjusted and unadjusted (see [12]) for factors that may have influenced performance within and across experiments. Unadjusted consistency included only experiment as a fixed effect. Adjusted consistency included experiment and also the following fixed effects that were identified as significant in the analyses described above and below: feeder position, malfunctioning time of own and other feeders, bias towards edge feeders (see above for description of bias measurements), bias towards neighbouring feeders and bias towards feeder assigned in previous experiment (reversal experiments only). If consistency estimates are high and similar for unadjusted and adjusted analyses, this implies that

our estimates of between-individual differences in performance are robust to these confounding effects and therefore more likely to reflect intrinsic differences in discrimination learning ability. We estimated consistency values both across all three experiments, and separately for the two reversal learning phases only. Great tits and blue tits were analysed separately.

# 3. Results

## 3.1. Summary of visits across experiments

A total of 409 individual great tits and blue tits of known sex visited the feeders during the initial learning experiment, with a total of 96 833 visits. The total numbers of individuals appearing in, participating in and meeting the learning criterion in the three different experiments are shown in table 1. The mean learning speed and mean number of visits per individual that met the learning criterion for each category of species, sex and age are given in electronic supplementary material, table S1.

## 3.2. Participation

Of all the individuals that were detected visiting at least once during the initial learning experiment, 58.7% of them visited enough to be considered as participating (table 1). For the initial learning experiment, there was a significant interaction between species and age on participation (estimate = 0.91, s.e. = 0.45, $z = 2.03$, $p = 0.043$). Great tits of different ages participated at similar levels, but juvenile blue tits were more likely to participate than adult blue tits (table 1). None of the other fixed effects significantly affected participation rates (electronic supplementary material, table S2). Two hundred and four of 221 individuals that had met the initial learning criterion continued to visit feeders and also participated in the first reversal experiment, and 183/198 individuals who learned during the previous two experiments continued to visit feeders and participated in the second reversal (table 1). We found no effects of age, sex or species on whether individuals that were detected at the feeder in the initial learning experiment ultimately met all of the criteria for participating and learning in all three experiments (electronic supplementary material, table S3).

## 3.3. Comparing learning criteria

We compared our chosen learning criterion of 80% correct visits over 20 consecutive visits to five other learning criteria that were based on two variables: the percentage of visits that were correct (either 80 or 90%) and the number of consecutive visits over which that percentage was calculated (10, 20 or 30 consecutive visits). These criteria were highly correlated with one another (range of correlation coefficients for initial learning: 0.69–0.96; first reversal learning: 0.68–0.95; second reversal learning: 0.76–0.98), indicating that birds met the different criteria in a similar number of visits. However, not all birds met all of the criteria. In initial learning, 221 individuals met the least stringent criterion (80% correct choices over 10 consecutive visits) compared with 205 individuals in the most stringent criterion (90% correct choices over 30 visits). Similar reductions in numbers reaching criterion were seen for the first reversal learning (200–183) and second reversal learning experiments (187–166).

For the chosen criterion of 80% correct choices over 20 consecutive visits, most individuals continued to perform above the 80% criterion on subsequent visits after having met the criterion (number of individuals choosing the correct feeder at least 80% of the time over all visits after those in which it met the learning criterion/number of individuals that met the learning criterion; initial learning: 166/221, first reversal learning: 132/198, second reversal learning: 124/183).

## 3.4. Factors affecting learning speeds

A summary of the findings of our analyses of factors affecting learning speeds is given in table 2.

In the initial learning experiment, great tits learned significantly faster than blue tits (electronic supplementary material, figure S2; estimate = −0.72, s.e. = 0.16, $t = -4.53$, $p < 0.001$). There was no significant effect of age, sex or any interactions of these variables with each other or with species on initial learning speed (electronic supplementary material, table S4). Individuals learned faster when they were assigned to feeders on the edge of the array than when they were assigned to the centre (figure 1$a$; estimate = −1.55, s.e. = 0.17, $t = -9.32$, $p < 0.001$). Individuals that experienced longer feeder outages took longer to learn, regardless of whether it was their own feeder, or another feeder

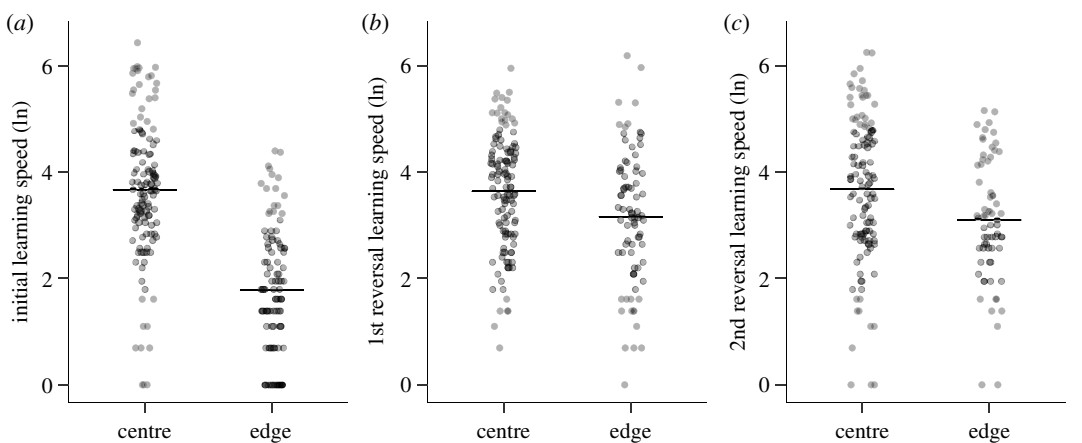

**Figure 1.** Effects of the location of the feeder on individual learning speed for (*a*) initial learning, (*b*) the first reversal and (*c*) the second reversal. Points represent individual birds (points have been jittered along the *x*-axis and rendered partially transparent to reduce overlap; as a result, any remaining overlap results in darker points). Horizontal line represents the mean value. Learning speed is the number of visits to criterion; therefore, lower values represent faster learning.

**Table 2.** Summary of factors influencing learning speed. Full results of statistical tests are given in electronic supplementary material, table S4. ✗ indicates no significant effect of the factor on learning speed. ✓ indicates there was a significant effect of the factor on learning speed.

|  | initial learning | first reversal | second reversal |
| --- | --- | --- | --- |
| age | ✗ | ✗ | ✗ |
| sex | ✗ | ✗ | ✗ |
| species | ✓[a] | ✗ | ✗ |
| feeder position | ✓[b] | ✓[b] | ✓[b] |
| inter-trial interval | ✗ | ✗ | ✗ |
| other feeders malfunction time | ✓[c] | ✓[c] | ✓[c] |
| own feeder malfunction time | ✓[c] | ✓[c] | ✓[c] |
| social group treatment | ✗ | ✗ | ✗ |

[a]Great tits learned faster than blue tits.
[b]Birds assigned to the edge learned faster than birds assigned to the centre.
[c]Birds took longer to learn when their assigned (or non-assigned) feeder was unpowered for a greater amount of time. Other factors that never had a significant effect and interactions are reported in electronic supplementary material, table S4.

(own feeder: estimate = 0.07, s.e. = 0.02, $t = 3.85$, $p < 0.001$; other feeders: estimate = 0.07, s.e. = 0.01, $t = 5.31$, $p < 0.001$). There was no effect of the average time interval between successive visits on learning speeds (electronic supplementary material, table S4).

In the first reversal learning experiment, there were again significant effects of feeder location (figure 1*b*; individuals assigned to the edge learned more quickly than individuals assigned to the centre; estimate = −0.43, s.e. = 0.16, $t = −2.70$, $p = 0.008$) and the duration of feeder outages on reversal learning speed (own feeder: estimate = 0.12, s.e. = 0.04, $t = 3.15$, $p = 0.002$; other feeders: estimate = 0.06, s.e. = 0.02, $t = 3.05$, $p = 0.003$). There was a trend towards an effect of the number of rewards obtained after learning in the initial learning experiment on first reversal learning speed (estimate = 0.001, s.e. = 0.0006, $t = 1.97$, $p = 0.050$); individuals that had obtained more rewards in the previous experiment learned more slowly during the reversal experiment. There were no effects of sex or age, as in the initial learning experiment. However, in contrast with the initial learning experiment, in the first reversal learning experiment, there was no effect of species on the reversal learning speed (electronic supplementary material, figure S2 and table S4). Social group treatment, learning speed in the previous experiment, and the average visit interval also had no significant effects on the first reversal learning speed (electronic supplementary material, table S4).

**Table 3.** Summary of the analyses of biases in errors. We tested for three types of bias in the types of errors that were made by individuals in our experiments. A detailed description of the analyses and statistics are given in the main text. Here, we summarize our findings by experiment and type of bias. We examined whether there was bias towards (i) edge or centre feeders, (ii) feeders that either neighboured or did not neighbour an individual's assigned feeder, and (iii) the feeder that the individual was assigned to in the previous experiment compared with feeders it had not been assigned to. For all bias types, we compared whether the bias was greater for great tits or blue tits, and for the first two we compared whether individuals assigned to a feeder in the centre were more biased than individuals assigned to a feeder on the edge. n.s., not significant; n.a., not applicable.

|  | initial | first reversal | second reversal |
|---|---|---|---|
| overall bias towards edge | edge > centre | n.s. | centre > edge |
| feeder assignment bias towards edge | n.s. | n.s. | n.s. |
| species bias towards edge | n.s. | n.s. | n.s. |
| overall bias towards neighbour | neighbour > non-neighbour | neighbour > non-neighbour | neighbour > non-neighbour |
| feeder assignment bias towards neighbour | n.s. | centre > edge | centre > edge |
| species bias towards neighbour | n.s. | n.s. | n.s. |
| overall bias towards assigned feeder of previous experiment | n.a. | assigned > non-assigned | assigned > non-assigned |
| species bias towards assigned feeder of previous experiment | n.a. | great tit > blue tit | great tit > blue tit |

Once again in the second reversal learning experiment, there were significant effects of feeder location (figure 1c; individuals assigned to the edge learned more quickly than individuals assigned to the centre; estimate = −0.59, s.e. = 0.17, $t$ = −3.50, $p < 0.001$) and the duration of feeder outages on reversal learning speed (own feeder: estimate = 0.12, s.e. = 0.02, $t$ = 5.35, $p < 0.001$; other feeders: estimate = 0.04, s.e. = 0.01, $t$ = 3.0, $p$ = 0.003). Individuals that had made more rewarded visits in the previous experiment (i.e. the first reversal learning) learned significantly more quickly in the second reversal learning experiment (estimate = −0.0003, s.e. = 0.0009, $t$ = −3.0, $p$ = 0.003). There were no significant effects of social group treatment, sex, age, species (electronic supplementary material, figure S2) or their interactions, or of the average time interval between visits or the learning speed in the first reversal learning experiment on learning speed in the second reversal learning experiment (electronic supplementary material, table S4).

## 3.5. Biases in errors

A summary of the findings of our analyses of bias in errors is given in table 3.

When individuals made an error in the initial learning experiment, they were more likely to make an error at a feeder located at the edge of the array than would be expected if their errors were distributed randomly (figure 2a; Wilcoxon test, $V$ = 14 750, $p < 0.001$, $N$ = 219 individuals). There was a trend for birds assigned to the centre to be more biased towards unassigned edge feeders (taking into account the different null probability for making an error at each feeder type depending on the type the individual was assigned to: figure 2a; WMW, $W$ = 6751, $p$ = 0.08, $N$ = 219). There was no difference between great tits and blue tits in the bias towards edge feeders (WMW, $W$ = 5625, $p$ = 0.76, $N$ = 219).

By contrast, in the first reversal learning experiment, there was no significant bias towards either edge or centre feeders (figure 2b; Wilcoxon test, $V$ = 9854, $p$ = 0.90, $N$ = 198). Individuals assigned to the centre did not differ in their bias towards unassigned edge feeders from individuals assigned to the edge (figure 2b; WMW, $W$ = 4952, $p$ = 0.56, $N$ = 198). However, individuals that had been assigned to the centre in the initial learning experiment had a stronger bias towards unassigned centre feeders in the first reversal learning experiment (WMW, $W$ = 1265, $p < 0.001$, $N$ = 198). There was no difference between great tits and blue tits in the bias towards edge feeders in this experiment (WMW, $W$ = 4478, $p$ = 0.73, $N$ = 198).

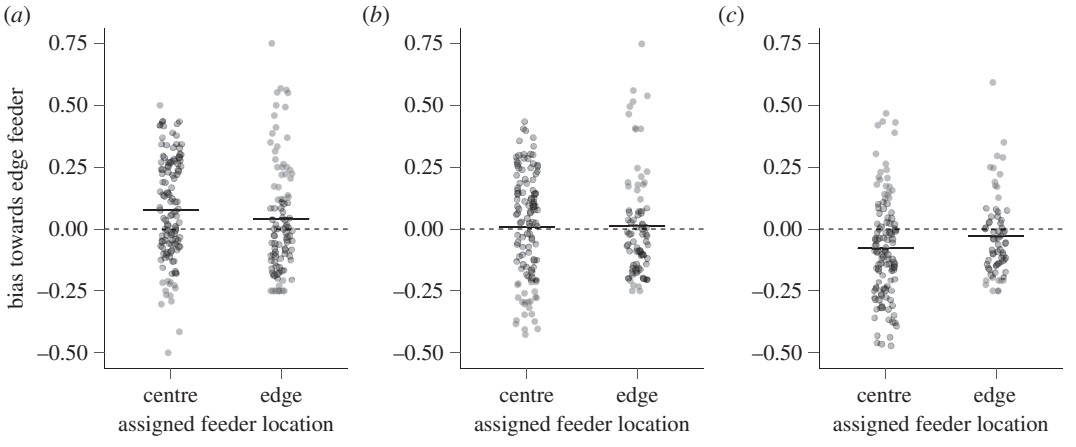

**Figure 2.** Bias in errors towards feeders located on the edge for (*a*) initial learning, (*b*) the first reversal and (*c*) the second reversal. The bias is calculated as the difference between the proportion of all error visits that were made on edge feeders and the proportion of such visits that would be expected if visit errors were distributed randomly with respect to feeder location. This expected proportion differs for individuals assigned to feeders in the centre (0.5) and individuals assigned to feeders on the edge (0.25). Points represent individual birds; solid horizontal lines represent mean values. The dotted horizontal line at $y = 0$ illustrates the null expected bias.

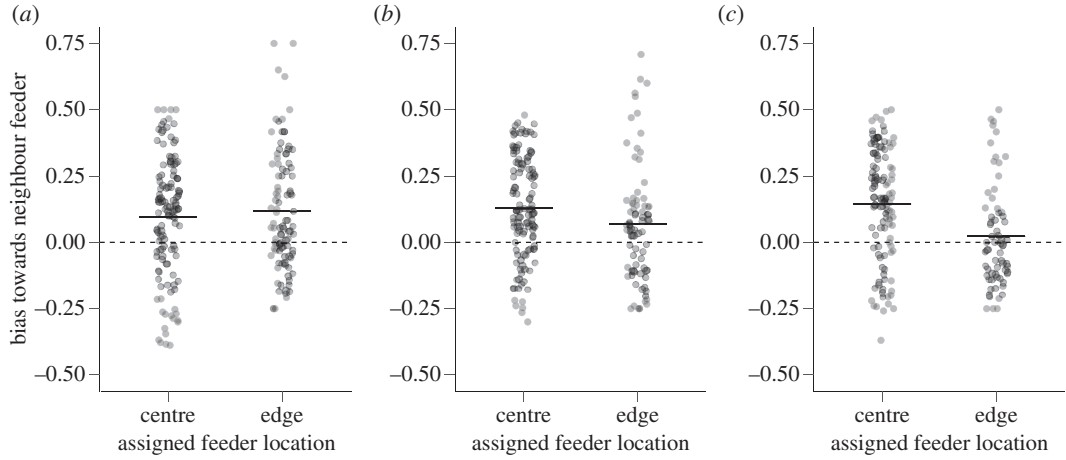

**Figure 3.** Bias in errors towards feeders neighbouring the bird's assigned feeder for (*a*) initial learning, (*b*) the first reversal and (*c*) the second reversal. The bias is calculated as the difference between the proportion of all error visits that were made on neighbouring feeders and the proportion of such visits that would be expected if visit errors were distributed randomly with respect to feeder location. This expected proportion differs for individuals assigned to feeders in the centre (0.5) and individuals assigned to feeders on the edge (0.25). Points represent individual birds; solid horizontal lines represent the mean values. The dotted horizontal line at $y = 0$ illustrates the null expected bias.

In the second reversal learning experiment, there was again a significant bias based on feeder location, but this time individuals were biased towards feeders in the centre rather than the edge (figure 2*c*; Wilcoxon test, $V = 5007$, $p < 0.001$, $N = 183$). There was a non-significant trend for birds assigned to centre feeders to be more biased towards centre feeders (figure 2*c*; WMW, $W = 3310$, $p = 0.095$, $N = 183$). Individuals that had been assigned to a centre feeder during the first reversal learning experiment had a stronger bias towards unassigned centre feeders in the second reversal learning experiment (WMW, $W = 1730$, $p < 0.01$, $N = 183$). There was no difference between great tits and blue tits in the bias towards centre feeders in this experiment (WMW, $W = 3934$, $p = 0.95$, $N = 183$).

Individuals in the initial learning experiment were more likely to make an error towards a neighbouring feeder than would be expected by chance (figure 3*a*; Wilcoxon test, $V = 17\,498$, $p < 0.001$, $N = 219$), but there was no difference between centre and edge birds in the bias towards neighbouring feeders (figure 3*a*; WMW, $W = 5904$, $p = 0.96$, $N = 219$). There was also no difference between great tits and blue tits in the bias towards neighbouring feeders (WMW, $W = 5187$, $p = 0.21$, $N = 219$).

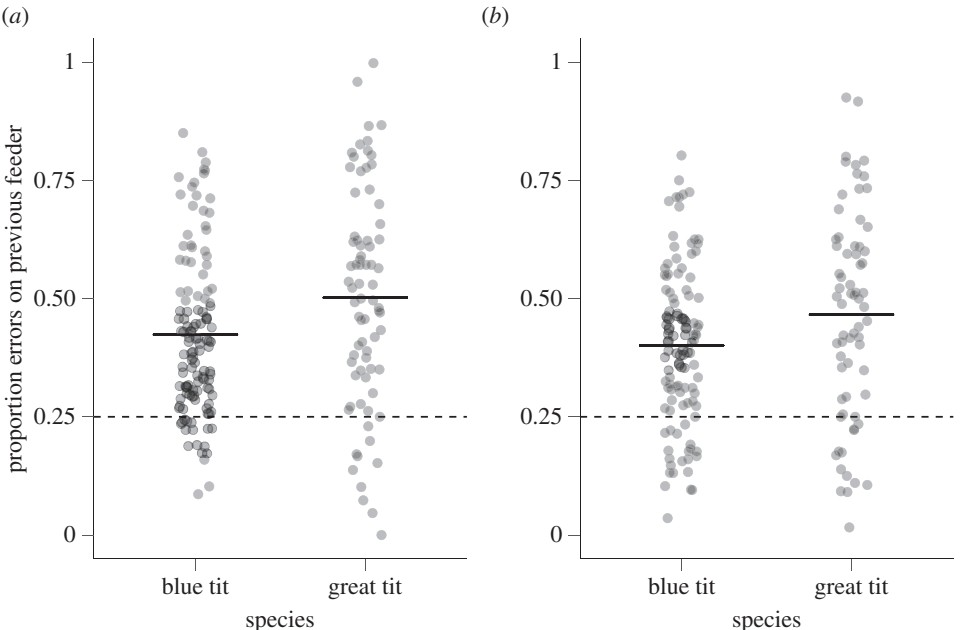

**Figure 4.** The proportion of errors made on the feeder that the individual had been rewarded by in the previous experiment for (*a*) the first reversal and (*b*) the second reversal. In contrast with figures 2 and 3, here we show raw values for error proportions and we show differences between the two species rather than the feeder location assignments. Points represent individual birds; solid horizontal line represents the mean value. The dotted horizontal line at $y = 0.25$ illustrates the null expectation if visit errors were distributed randomly with respect to the identity of the previously assigned feeder.

There was again an overall bias in making errors on feeders neighbouring the assigned feeder in the first reversal learning experiment (figure 3*b*; Wilcoxon test, $V = 14\,550$, $p < 0.001$, $N = 198$). However, in contrast with the initial learning experiment, here individuals assigned to the centre were more biased towards making errors on neighbouring feeders than were individuals assigned to the edge (figure 3*b*; WMW, $W = 5692$, $p = 0.014$, $N = 198$). There was no difference between great tits and blue tits in the bias towards neighbouring feeders (WMW, $W = 4366$, $p = 0.53$, $N = 198$).

Individuals in the second reversal learning experiment were again biased towards neighbouring feeders (figure 3*c*; Wilcoxon test, $V = 11\,986$, $p < 0.001$, $N = 183$), and as in the first reversal learning experiment, individuals assigned to the centre were more biased towards making errors on neighbouring feeders than were individuals assigned to the edge (figure 3*c*; WMW, $W = 5243$, $p < 0.001$, $N = 183$). There was no difference between great tits and blue tits in the bias towards neighbouring feeders (WMW, $W = 3983$, $p = 0.83$, $N = 183$).

There was a significant bias in the first reversal learning experiment towards making errors on the feeder to which the bird had been assigned during the initial learning experiment (Wilcoxon test, $V = 18\,298$, $p < 0.001$, $N = 198$). Great tits were significantly more biased than blue tits towards errors on the feeder to which they had been previously assigned (figure 4*a*; WMW, $W = 3543$, $p = 0.006$, $N = 198$). In the second reversal learning experiment, there was again a significant bias towards making errors on the feeder to which the bird had been assigned during the first reversal learning experiment (Wilcoxon test, $V = 14\,782$, $p < 0.001$, $N = 183$), and again this bias was stronger for great tits than it was for blue tits (figure 4*b*; WMW, $W = 3187$, $p = 0.037$, $N = 183$).

## 3.6. Individual consistency in learning speed

Consistency measures were very similar whether or not they were adjusted for several fixed effects (see Methods). For simplicity, we report the adjusted values here; unadjusted values are given in electronic supplementary material, table S5. Great tits did not show significant consistency in learning speed when all three experiments were considered ($R = 0.009$, s.e. $= 0.05$, $p = 0.5$). By contrast, individual reversal learning speeds were consistent in great tits across the two reversal experiments (figure 5*a*; $R = 0.28$, s.e. $= 0.11$, $p = 0.01$); individuals that were faster in the first reversal were also faster in the second reversal. For blue tits, there was a trend towards statistically significant consistency in learning speed when all three experiments were considered ($R = 0.09$, s.e. $= 0.06$, $p = 0.06$), but no significant

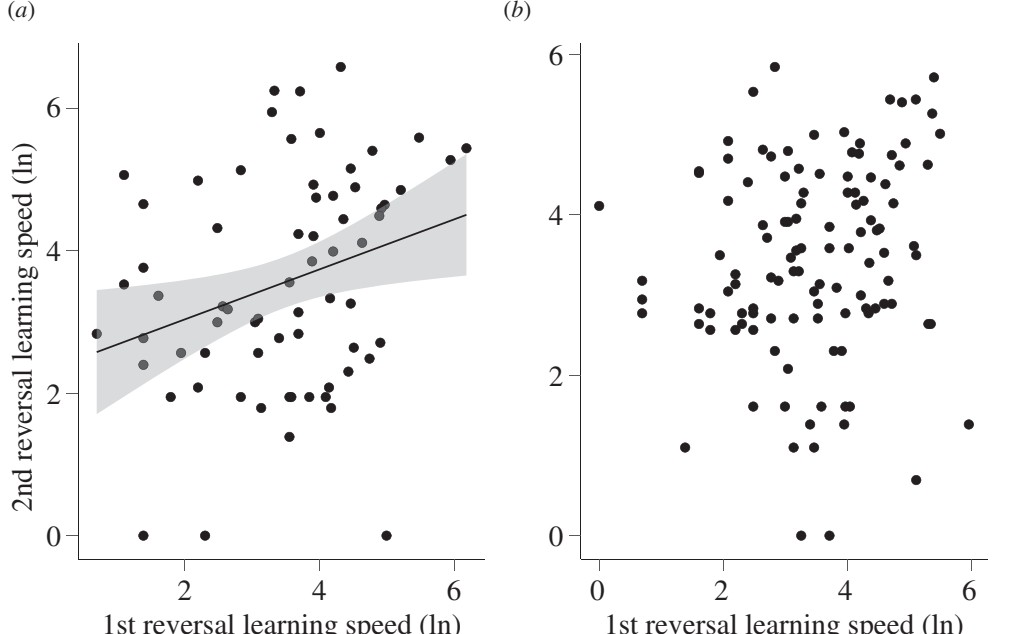

**Figure 5.** Consistency in individual variation in (natural log-transformed) learning speed in the first and second reversal experiments for (a) great tits and (b) blue tits. Each individual is illustrated by a separate point. A regression line with 95% confidence interval is included for great tits because there was significant consistency in learning speeds in the two reversal experiments (note that consistency was not calculated using a linear regression; see text). No line is shown for blue tits because they did not have significant consistency in learning speeds.

consistency when just the two reversal experiments were considered (figure 5b; $R = 0.12$, s.e. $= 0.09$, $p = 0.13$); the value of the coefficient was relatively low in both cases.

## 4. Discussion

Using a large-scale experiment in the wild, we examined multiple factors that determined individual-level participation and performance in a discrimination learning and two reversal learning tasks, in two songbird species. The largest bottleneck was identified at the level of participation in the initial learning experiment; once birds had participated, the majority of them went on to reach the learning criterion in all three experiments. The demographic factors of age and species affected participation and learning in some experiments, but sex had no effects on participation or learning. We also explored the dynamics of learning by examining whether there were patterns in the types of mistakes that individuals made while searching for their rewarding feeder, and indeed, we found several biases related to the location of the feeder in the array and the individual's previous experience of reward. Reversal learning performance was consistent within individuals in great tits, suggesting that despite many potential confounds, we are able to capture meaningful variation in cognitive ability in this task for this species.

### 4.1. Who and what proportion of the population participates?

In field studies generally, it is important to determine whether there are biases in patterns of participation that could affect interpretation (see also [61]). Here, we found that only 58.7% of the birds that were detected on the feeders during the initial learning experiment made enough visits to be considered as participants. However, few of the variables that we explored had a significant effect on the likelihood of participation, or indeed on whether birds ultimately passed all of the participation and learning criteria for all three experiments. Furthermore, our participation rates were generally much higher than those reported from other cognition studies on wild populations, in which rates of full participation among individuals that interact with tasks are usually less than 50% and in many cases less than 10% (reviewed by van Horik et al. [18]). Indeed, similar previous studies of cognitive performance in this same population also found that some species, including blue tits, failed to participate in large numbers [30,33], perhaps because of

reduced ability to interact with devices due to smaller body size. A major advantage of our cognitive apparatus is that it required almost no physical manipulation (push lightweight, transparent door) to access the reward. This reduces the confounding effects of physical ability or motor biases that have proven problematic in other studies, and facilitates cognition studies in mixed-species flocks [9]. Nevertheless, we caution that our calculation of participation rates only includes those birds that were detected visiting a feeder. Rates of participation tend to be even lower when including all individuals that potentially could have visited the device but for unknown reasons did not [18]. On the other hand, many of the individuals that were classified as non-participants may have only occasionally passed through the area, and therefore were not exposed to the task for very long. To obtain a fuller picture of the factors influencing participation rates and their consequences for estimates of variation in cognitive performance, devising new methods to characterize movements and other behaviours of non-participating individuals is needed.

## 4.2. Confounding effects on learning speed

We found no evidence for an effect of sex or age on cognitive performance. Our results on lack of influence of sex are consistent with previous studies in great tits in this same population [34,62,63], even though in a laboratory study of observational learning of the location of food caches, female great tits learned faster than males [64]. Meanwhile, several studies of great tits found that juveniles were more likely to learn or learned more quickly than adults in colour association and problem-solving tasks (although these effects were often weak and not apparent in all years) [33,34,62,63]. By contrast, we found no effect of age on discrimination learning. Effects of sex or age on cognitive performance are highly variable across studies, indicating that these effects are dependent on species, task or context [14–17].

Even though great tits learned faster than blue tits in the initial learning experiment, there were no differences between the two species in reversal learning speeds. This difference between experiments arose primarily because great tits learned quickly in the initial learning experiment but took longer to learn in the reversal experiments, whereas blue tits' learning speeds were generally similar across all three experiments (electronic supplementary material, figure S2). There are two potential explanations for species differences in initial learning performance: (i) these species inherently differ in cognitive ability, and (ii) these species do not differ in cognitive ability but instead differ only in cognitive performance (i.e. the observed performance on a task at a given time in a given context). Differences between the species in cognitive performance could arise, for instance, because great tits are larger and superior competitors to blue tits [65,66]. However, understanding how competition influences cognitive performance requires further investigation, as blue tits may either be displaced from their rewarded feeder or, alternatively, may be slow to learn because their attention is focused on avoiding displacements from competitors. If competition is an important factor, we would expect learning speed to improve if only one species had access to the feeders at a time. However, such an experimental design is not informative for studying cognitive ecology in the wild because these species normally forage in mixed-species flocks. Thus, cognitive performance, not ability, is likely to be the more relevant variable for understanding variation in cognition in the wild, not least because selection is more likely to act directly on performance traits than underlying mechanisms (see [67]). However, if competition from great tits explained poorer cognitive performance in blue tits, then we would have expected to see species differences across all of the experiments. Species differences were not present in the reversal learning phases, suggesting either that species differences may be task-specific or that the effect of competition was only apparent during initial learning when the task was novel and more challenging.

In addition to individual attributes such as age, sex and species, the spatial position of the feeders relative to one another may have influenced performance. Indeed, birds assigned to feeders on the edge of the array learned faster in all three experiments, possibly because detecting and discriminating a feeder on the edge is easier than discriminating among multiple feeders in the centre of the array. However, these results could also be explained by factors unrelated to the ease of the discrimination task. For instance, interference with other individuals may have been less pronounced on the edge than in the centre. Furthermore, in the initial learning experiment, there was a general bias of visits towards feeders on the edge (see below), and this bias could have made it easier for birds to learn edge feeders if they were assigned to one (although we note that there was no statistically significant difference in the proportion of errors made at edge feeders between individuals assigned to the edge and those assigned to the centre; figure 2). However, this explanation does not hold for either of the two reversal experiments. Indeed, in the second reversal, birds assigned to the edge continued to learn faster despite an overall bias towards making errors on feeders in the centre (figure 2c). The quantification of biases in

patterns of visitation is important for understanding the dynamics of learning, but we note that these biases did not affect consistency in performance (see below), indicating that our estimates of individual variation were robust to the types of errors made while learning.

As field-based technologies allow experiments to move into an automated era (just as tracking devices have allowed a move into automated observation), it is particularly important to consider the effects of device and automation reliability. In our study, the amount of time during which the feeders malfunctioned and were inaccessible could have biased estimates of learning speed in two ways. First, if an individual's assigned feeder malfunctioned, this would have disrupted the formation of a learned association between reward and feeder location. Second, if feeders not assigned to the individual malfunctioned, we would have missed any erroneous visits made to that feeder, and thus inferred a faster learning speed than we should have. We believe that the non-assigned feeder malfunctions were not a major source of bias because of the positive relationship between the duration of non-assigned feeder malfunction and learning speed (i.e. individuals that experienced more non-assigned malfunctions learned more slowly). Nevertheless, malfunction at the assigned feeder probably slowed down learning for some individuals. We attempted to control for this statistically, but acknowledge that the estimates of learning speed for some individuals potentially indicate a lower cognitive performance than is actually the case. This is because slow learners were disproportionately affected since fast learners were more likely to meet the learning criterion before any feeder malfunction took place.

## 4.3. Dynamics of learning and errors

We examined temporal, spatial and experiential effects on learning speed as well as on the pattern of visits to gain insights into how individuals learned the task. Typical experimental designs for studies of learning in the laboratory are able to control for variables such as inter-trial intervals and reward experience by setting specific criteria before advancing subjects to subsequent test phases on a regular schedule. These controls are not always possible in field experiments. Although we did not control the amount of time between an individual's visits, we found no evidence that this variable impacted performance in our study: individuals whose visits were more distantly spaced in time did not meet the learning criterion in fewer total visits, as would be predicted from previous theoretical and laboratory studies of learning and memory (reviewed by the authors in [59,60]). However, we did show that reward history after the learning criterion was reached influenced learning in the reversal experiments, an effect that can be controlled for statistically or alternatively by implementing individual-specific automated programmes [33].

As expected for tasks in which reward contingencies have been switched, we found that birds were more likely to make errors on their previously rewarded feeder in the reversal phases. Our experimental set-up consisted of five choices (as opposed to a binary choice), allowing us to explicitly investigate additional sources of bias that may affect error rate. We showed that visits were biased by the spatial position of the feeders with respect to one another. Errors were more likely to be made on feeders that neighboured the assigned feeder. In both reversal learning experiments, this bias was stronger for birds assigned to the centre than birds assigned to the edge. This suggests that individuals had the most difficulty in discriminating between feeders that had feeders located on either side, and supports our finding that birds assigned to the edge learned more quickly. However, we also found there were general spatial biases for edge versus central feeders, but that the direction of this bias shifted across the experiments, regardless of the location of individuals' assigned feeders. The pattern of increased bias towards centre feeders across the experiments may be explained by more individuals having experienced being rewarded previously by a centre feeder with each new experiment because centre feeders outnumber edge feeders. Overall, these results show that fine-scale spatial patterns probably influence cognitive performance.

## 4.4. Consistency and individual differences

Individuals that participated almost always learned, but there was a great deal of individual variation in learning speed. Some of these differences were consistent, pointing to intrinsic differences in cognitive performance; namely, in great tits, between-individual differences in performance were consistent for the two reversal experiments (figure 5a). Estimates for between-individual differences in behaviour are often affected by whether or not additional variables are controlled for [11]. However, we found that consistency estimates adjusted for potential confounding variables were very similar to unadjusted values, suggesting that our measurements of performance were robust to a variety of environmental

conditions. Demonstrating that consistent between-individual differences are an accurate measure of a clearly defined trait is an important prerequisite for studies on the evolution of cognitive performance in wild populations [2]. By contrast, blue tits were not consistent in their reversal learning performance. Although individual differences may simply be less pronounced in the blue tit, it may also be that individual differences may be masked in the social environment, and in particular by the effect of living in groups with larger and more dominant species such as the great tit. Whatever the mechanism, the difference in consistency shows that selection potentially will act on cognitive performance differently for species in a mixed-species environment.

Despite the evidence for consistency across the two reversal learning experiments in one species, we found only very low values for consistency across all three experiments for both species; in other words, an individual's learning speed in the initial learning experiment was unrelated to its learning speed in the reversal learning experiments. A positive relationship between performance in these tasks is predicted if there is some underlying cognitive factor that affects performance in multiple contexts [44,45]. Alternatively, trade-offs between initial learning and reversal learning performance are frequently postulated because of a trade-off between speed and flexibility [42,43]. Neither of these hypotheses was supported by our data, suggesting the two traits are independent. This result is not entirely unexpected because different cognitive abilities and brain regions are thought to be involved in these two tasks [40,41], and some previous studies have also reported that initial and reversal performance are not correlated [68,69]. Nevertheless, as outlined above, many studies find covariation in initial and reversal learning, and we cannot discount the possibility that some important confounds (e.g. neophobia [70,71]) were not controlled for, obscuring relationships between these two behaviours in our study system.

## 5. Conclusion

Both individual-level and local environmental scale factors affected cognitive performance in our experiment, as in previous studies of cognition in the wild [18,63,72,73]. Intuitively, it should be expected that a multitude of factors influence cognitive performance, both because cognitive ability itself is underpinned by diverse neurophysiological mechanisms [74], and because the resulting cognitive performance, like the expression of most behaviours, is highly dependent on context [75]. Our findings of consistent individual variation in some cases, along with our ability to rapidly measure learning speeds in large numbers of individuals, while accounting for several usually overlooked confounding variables, point the way towards the potential for advancing our understanding of cognition in wild animals.

Ethics. The research project received ethical approval from the Animal Welfare Body at University College Cork (HPRA licence number AE19130-P017), and was in accordance with the ASAB (Association for the Study of Animal Behaviour) Guidelines for the Treatment of Animals in Behavioural Research and Teaching. All research was conducted under BTO licences as part of ongoing research in this population.

Data accessibility. Our data are deposited at the Dryad Digital Repository: https://doi.org/10.5061/dryad.djh9w0vw5 [76].

Authors' contributions. J.L.Q., M.S.R., J.A.F., G.L.D. and I.G.K. designed the experiment, M.S.R., G.L.D. and I.G.K. built experimental devices, M.S.R. and S.J.C. performed fieldwork, M.S.R. analysed data and wrote manuscript with input from all authors, and all authors gave final approval for publication. Note that middle authors on this manuscript are listed in alphabetical order by surname.

Competing interests. The authors declare no competing interests.

Funding. Funding for M.S.R., S.J.C., G.L.D. and I.G.K. from the European Research Council under the European Union's Horizon 2020 Programme (FP7/2007-2013)/ERC Consolidator Grant 'EVOLECOCOG' Project No. 617509, awarded to J.L.Q., and by a Science Foundation Ireland ERC Support Grant 14/ERC/B3118 to J.L.Q.

Acknowledgements. Keith McMahon assisted with bird ringing and fieldwork. Martin Whitaker helped design the selective feeders, and James Savage and Iván de la Hera Fernández helped with their construction. Karen Cogan assisted with equipment sourcing and purchasing. Ben Sheldon provided access to the study system at Wytham Woods and funding to maintain the long-term study. We thank Joah Madden and two anonymous reviewers for providing helpful comments, and Theresa Burt, Sue Healy and Vladimir Pravosudov for useful discussion.

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
