## [Reviewer comments · Royal Society Open Science]

Review History

RSOS-192107.R0 (Original submission)

Review form: Reviewer 1 (Joah Madden)

Is the manuscript scientifically sound in its present form?

Yes

Are the interpretations and conclusions justified by the results?

Yes

Is the language acceptable?

Yes

Do you have any ethical concerns with this paper?

No

Have you any concerns about statistical analyses in this paper?

No

Recommendation?

Accept with minor revision (please list in comments)

Comments to the Author(s)

Please see attached file (Appendix A).

Review form: Reviewer 2**Is the manuscript scientifically sound in its present form?**

Yes

Are the interpretations and conclusions justified by the results?

Yes

Is the language acceptable?

Yes

Do you have any ethical concerns with this paper?

No

Have you any concerns about statistical analyses in this paper?

No

Recommendation?

Accept with minor revision (please list in comments)

Comments to the Author(s)

I reviewed the manuscript "Determinants of individual variation in discrimination learning performance in wild mixed species flocks" for potential publication in Royal Society Open Science. The authors report some behavioral and demographic variables that affected performance on field-tested discrimination learning and reversal learning tasks in a mixed-species population. In my opinion, the presented data is scientifically sound and represent a valuable contribution to the field of behavioral ecology. Globally, I found the behavioral protocol and statistical analyses appropriate and in accordance with the standards in the field. I particularly liked the fact that they succeeded in testing a considerable sample size using a 'classical' discrimination learning task that normally requires a large time investment and which makes it rarely possible to test such high numbers.

That being said, I do have some minor and some less minor concerns about a few aspects of the manuscript and analyses. I detail them in chronological order of the manuscript below.

Introduction

L51-62: I suggest to completely remove this section on the effect of participation on cognitive measurements. First, participation is expected to have a strong effect on the performance in problem-solving tasks, in which the number of interactions with the task inevitably increases chances of success. Most of the cited references (including [15], which is not a review) are in fact studies that examined, at least in part, problem-solving. For discrimination learning, I do not think that this is a "major source of bias" as suggested by the authors. It may have an effect, possibly through the mechanisms suggested at the end of the paragraph, but I do not think that it's very likely and it's somewhat speculative. Finally, as I will explain in detail below, I believe that there is insufficient data on participation in this study to draw any meaningful conclusion concerning its effect on learning performance. The participation rates (but not in models, see

below) can still be presented in this manuscript as an information on the actual sample size, however, much less focus should be put on this aspect in the introduction, results and discussion.

L80, and throughout the rest of the manuscript: I am puzzled as to why this experiment is not qualified as “spatial learning”? There are a few different ways to measure spatial learning, and the method described here appears to be one of them to my knowledge. If there is a good reason to avoid using the term spatial, I would suggest mentioning it in the manuscript to help the reader understand why this choice was made.

L82: I suggest to remove “participation”.

L67: It would increase the credibility of this sentence if one or more references that investigated a more similar model/context were added.

L90-92: The cited reference does not address the question of the effect on discrimination learning on survival and reproduction. There is in fact no mention of learning or any other cognitive trait in this article. I therefore suggest remove it. Unless other references are found to support it, this sentence should perhaps also be removed.

L99-100: Without further explanations, it is impossible to understand what this sentence refers to. I would suggest either removing it, or providing a little more information so that the reader has at least an idea of what this is about.

L100-102: The “consistency” effect is problematic since it is measured using 3 different assays (even if they are measured using the same apparatus) that are normally assumed to measure different cognitive traits. Moreover, assuming that “consistent” measurements would suggest that those traits are targets of selection is very speculative. I suggest eliminating this analysis and interpretation. On the other hand, examining the relation between the three learning phases using appropriate models (e.g. linear regression) is indicated.

Methods

L146-149: If the data is available, I would be interesting to test the effect of the distance between the birds’ initial capture site and the experiment location on the learning performance.

L229-231: How was the minimum 50 trials to participation chosen? Why not simply retain all the birds that reached the learning criterion? And why was the 1-bird exception retained?

L257: I recommend to remove “This assumption is likely valid”, it is not necessary here.

L287-293: I strongly believe that all the animals that had dysfunctional apparatuses should be removed from the analyses. It is safe to assume that a non-consistent opening of the correct feeder will have a considerable effect on learning speed. The strength of this effect could vary depending on the time when it happened (the trial that the bird had reached when this occurred, or any other variable that could have had an effect on learning when the feeder malfunctioned). I think that such a fundamental flaw in the behavioral protocol cannot be “corrected for” and therefore all those animals should be discarded.

L331-334: As pointed out above, I do not think that this analysis is appropriate. If different traits are being measured (as acknowledged by the authors), then there is no such thing as “consistency”. Furthermore, the performance across tasks should not be compared using a repeatability tool. It would be more appropriate to assess the relation between the performance in each assays using correlations and/or linear regressions.

Results

Table 1: I do not think that this table is necessary as a main table. It could be added in supplementary information, or alternatively combined with another, more relevant, table.

L368-373: If almost all birds that learned in the initial learning phase also participated in reversal learning, then I am not sure that it is relevant to try to explain this “variation”. There is an almost inexistent variation here (n=14/409 birds that participated in RL but not IL) so I doubt that this statistical analysis is sound. I suggest to remove it.

L382-397: I do not think that all this section is necessary.

L404-407: As per above comment, I would remove this.

L418: I suggest to replace “but, in the first...” by “in contrast, there was no effect of the species on the reversal learning speed”.

L419-420: The result that reversal learning speed was not related to initial learning is extremely surprising. There is very often a strong correlation between those measurements. I believe that this unexpected result should be more discussed in the discussion section.

L424: As per the above comment, I would remove the birds that experienced outages, therefore this variable could be removed from models.

L440: I would recommend rearranging this section (by bias type instead of by learning type?) and maybe adding subtitles to help the reader follow this section. The different types of biases are easily confused with each other. I had to go through this section numerous times to fully understand what the results were. Maybe a table summarizing all those results would help. In fact, a table might be more useful than the Figures 2 and 3.

L467-471: I would not denote the visits to previously learned feeder a “bias”. It’s the very logic of this experiment and it is expected that the birds will visit the previously learned feeder first (retention phase), before eventually figuring out that it is not rewarded anymore. I suggest to reword.

Figure 5: Is this result obtained by the “repeatability” analysis? If yes, then it is misleading as the graph represents a regression (and as stated in the legend).

L534: Innovation and discrimination learning are two distinct traits, therefore I do not think that comparing the demographic factors that affect them is appropriate here. I suggest removing this sentence or rewording it.

L586-587: Despite the absence of statistical difference between the species’ performance on RL1 and RL2, there is a pronounced shift in the species performance from IL to RL2. Great tits are better in IL but blue tits are better in RL2. They actually have opposite slopes throughout the stages, blue tits have a negative slope and great tits have a positive slope. I think that this is very interesting and that it should be analyzed using appropriate statistical analyses (comparing slopes and/or the difference between RL2 and IL).

L623-636: This section could be removed if the birds that experienced malfunctions were discarded.

L649-651: I suggest to remove this sentence as it is very speculative, and I think that it is not suitable to use the word “brightest” in this context.

Decision letter (RSOS-192107.R0)

24-Feb-2020

Dear Professor Reichert

On behalf of the Editors, I am pleased to inform you that your Manuscript RSOS-192107 entitled "Determinants of individual variation in discrimination learning performance in wild mixed species flocks" has been accepted for publication in Royal Society Open Science subject to minor revision in accordance with the referee suggestions. Please find the referees' comments at the end of this email.

The reviewers and handling editors have recommended publication, but also suggest some minor revisions to your manuscript. Therefore, I invite you to respond to the comments and revise your manuscript.

- Ethics statement

- Data accessibility

<http://datadryad.org/submit?journalID=RSOS&manu=RSOS-192107>

- Competing interests

- Authors' contributions

AB carried out the molecular lab work, participated in data analysis, carried out sequence alignments, participated in the design of the study and drafted the manuscript; CD carried out the statistical analyses; EF collected field data; GH conceived of the study, designed the study,

coordinated the study and helped draft the manuscript. All authors gave final approval for publication.

- Acknowledgements

- Funding statement

Because the schedule for publication is very tight, it is a condition of publication that you submit the revised version of your manuscript before 04-Mar-2020. Please note that the revision deadline will expire at 00.00am on this date. If you do not think you will be able to meet this date please let me know immediately.

If your manuscript is newly submitted and subsequently accepted for publication, you will be asked to pay the article processing charge, unless you request a waiver and this is approved by Royal Society Publishing. You can find out more about the charges at <https://royalsocietypublishing.org/rsos/charges>. Should you have any queries, please contact openscience@royalsociety.org.

on behalf of Dr Alecia Carter (Associate Editor) and Kevin Padian (Subject Editor)
openscience@royalsociety.org

Associate Editor Comments to Author (Dr Alecia Carter):
Comments to the Author:

Dear authors, I have now received two constructive reviews of your manuscript, which you will find below. Both reviewers found the study to be well executed and interesting. I also found the study to be interesting, and the approach to be rigorous and thorough. In general, the suggested edits should improve the digestibility of the manuscript for readers.

I disagreed with the reviewers on one point, however: the priority and inclusion of the analysis of task participation. I agree with the authors that it is overlooked in studies of cognition and warrants investigation. It is not tangential – if there is low participation of a particular group of individuals in a cognitive task, the ‘remaining’ results on cognition will be biased. I also feel that, if an analysis is deemed necessary before the data come in, then it should be kept i.e. there should be no post-hoc changes in analyses to accommodate a ‘tidier’ story. That’s not necessarily the case here, as it is true that the sample of non-participants in the reversal-learning trials may be too small to detect an effect, as highlighted by both reviewers. But for the sake of analysis integrity, I would encourage the authors to be explicit in their revision about their initial analysis plans and their subsequent decisions.

(One question that remains is whether this measure of participation already misses some birds i.e. those tagged and alive but who don’t go to the feeders, as acknowledged in the discussion by the authors, albeit briefly. Can’t these data be analysed, post-hoc? Of course, the missing birds

could be dead, but it still may have been useful information to include in this section of the MS, since the authors bring it up.)

Reviewer comments to Author:

Reviewer: 1

Comments to the Author(s)

Please see attached file (Reichart et al Review.pdf)

Reviewer: 2

Comments to the Author(s)

I reviewed the manuscript “Determinants of individual variation in discrimination learning performance in wild mixed species flocks” for potential publication in Royal Society Open Science. The authors report some behavioral and demographic variables that affected performance on field-tested discrimination learning and reversal learning tasks in a mixed-species population. In my opinion, the presented data is scientifically sound and represent a valuable contribution to the field of behavioral ecology. Globally, I found the behavioral protocol and statistical analyses appropriate and in accordance with the standards in the field. I particularly liked the fact that they succeeded in testing a considerable sample size using a ‘classical’ discrimination learning task that normally requires a large time investment and which makes it rarely possible to test such high numbers.

That being said, I do have some minor and some less minor concerns about a few aspects of the manuscript and analyses. I detail them in chronological order of the manuscript below.

Introduction

L51-62: I suggest to completely remove this section on the effect of participation on cognitive measurements. First, participation is expected to have a strong effect on the performance in problem-solving tasks, in which the number of interactions with the task inevitably increases chances of success. Most of the cited references (including [15], which is not a review) are in fact studies that examined, at least in part, problem-solving. For discrimination learning, I do not think that this is a “major source of bias” as suggested by the authors. It may have an effect, possibly through the mechanisms suggested at the end of the paragraph, but I do not think that it’s very likely and it’s somewhat speculative. Finally, as I will explain in detail below, I believe that there is insufficient data on participation in this study to draw any meaningful conclusion concerning its effect on learning performance. The participation rates (but not in models, see below) can still be presented in this manuscript as an information on the actual sample size, however, much less focus should be put on this aspect in the introduction, results and discussion.

L80, and throughout the rest of the manuscript: I am puzzled as to why this experiment is not qualified as “spatial learning”? There are a few different ways to measure spatial learning, and the method described here appears to be one of them to my knowledge. If there is a good reason to avoid using the term spatial, I would suggest mentioning it in the manuscript to help the reader understand why this choice was made.

L82: I suggest to remove “participation”.

L67: It would increase the credibility of this sentence if one or more references that investigated a more similar model/context were added.

L90-92: The cited reference does not address the question of the effect on discrimination learning on survival and reproduction. There is in fact no mention of learning or any other cognitive trait

in this article. I therefore suggest remove it. Unless other references are found to support it, this sentence should perhaps also be removed.

L99-100: Without further explanations, it is impossible to understand what this sentence refers to. I would suggest either removing it, or providing a little more information so that the reader has at least an idea of what this is about.

L100-102: The “consistency” effect is problematic since it is measured using 3 different assays (even if they are measured using the same apparatus) that are normally assumed to measure different cognitive traits. Moreover, assuming that “consistent” measurements would suggest that those traits are targets of selection is very speculative. I suggest eliminating this analysis and interpretation. On the other hand, examining the relation between the three learning phases using appropriate models (e.g. linear regression) is indicated.

Methods

L146-149: If the data is available, I would be interesting to test the effect of the distance between the birds’ initial capture site and the experiment location on the learning performance.

L229-231: How was the minimum 50 trials to participation chosen? Why not simply retain all the birds that reached the learning criterion? And why was the 1-bird exception retained?

L257: I recommend to remove “This assumption is likely valid”, it is not necessary here.

L287-293: I strongly believe that all the animals that had dysfunctional apparatuses should be removed from the analyses. It is safe to assume that a non-consistent opening of the correct feeder will have a considerable effect on learning speed. The strength of this effect could vary depending on the time when it happened (the trial that the bird had reached when this occurred, or any other variable that could have had an effect on learning when the feeder malfunctioned). I think that such a fundamental flaw in the behavioral protocol cannot be “corrected for” and therefore all those animals should be discarded.

L331-334: As pointed out above, I do not think that this analysis is appropriate. If different traits are being measured (as acknowledged by the authors), then there is no such thing as “consistency”. Furthermore, the performance across tasks should not be compared using a repeatability tool. It would be more appropriate to assess the relation between the performance in each assays using correlations and/or linear regressions.

Results

Table 1: I do not think that this table is necessary as a main table. It could be added in supplementary information, or alternatively combined with another, more relevant, table.

L368-373: If almost all birds that learned in the initial learning phase also participated in reversal learning, then I am not sure that it is relevant to try to explain this “variation”. There is an almost inexistent variation here (n=14/409 birds that participated in RL but not IL) so I doubt that this statistical analysis is sound. I suggest to remove it.

L382-397: I do not think that all this section is necessary.

L404-407: As per above comment, I would remove this.

L418: I suggest to replace “but, in the first...” by “in contrast, there was no effect of the species on the reversal learning speed”.

L419-420: The result that reversal learning speed was not related to initial learning is extremely

surprising. There is very often a strong correlation between those measurements. I believe that this unexpected result should be more discussed in the discussion section.

L424: As per the above comment, I would remove the birds that experienced outages, therefore this variable could be removed from models.

L440: I would recommend rearranging this section (by bias type instead of by learning type?) and maybe adding subtitles to help the reader follow this section. The different types of biases are easily confused with each other. I had to go through this section numerous times to fully understand what the results were. Maybe a table summarizing all those results would help. In fact, a table might be more useful than the Figures 2 and 3.

L467-471: I would not denote the visits to previously learned feeder a "bias". It's the very logic of this experiment and it is expected that the birds will visit the previously learned feeder first (retention phase), before eventually figuring out that it is not rewarded anymore. I suggest to reword.

Figure 5: Is this result obtained by the "repeatability" analysis? If yes, then it is misleading as the graph represents a regression (and as stated in the legend).

L534: Innovation and discrimination learning are two distinct traits, therefore I do not think that comparing the demographic factors that affect them is appropriate here. I suggest removing this sentence or rewording it.

L586-587: Despite the absence of statistical difference between the species' performance on RL1 and RL2, there is a pronounced shift in the species performance from IL to RL2. Great tits are better in IL but blue tits are better in RL2. They actually have opposite slopes throughout the stages, blue tits have a negative slope and great tits have a positive slope. I think that this is very interesting and that it should be analyzed using appropriate statistical analyses (comparing slopes and/or the difference between RL2 and IL).

L623-636: This section could be removed if the birds that experienced malfunctions were discarded.

L649-651: I suggest to remove this sentence as it is very speculative, and I think that it is not suitable to use the word "brightest" in this context.

Author's Response to Decision Letter for (RSOS-192107.R0)

See Appendix B.

Decision letter (RSOS-192107.R1)

23-Mar-2020

Dear Professor Reichert,

It is a pleasure to accept your manuscript entitled "Multiple factors affect discrimination learning performance, but not between-individual variation, in wild mixed-species flocks of birds" in its current form for publication in Royal Society Open Science.

on behalf of Dr Alecia Carter (Associate Editor) and Kevin Padian (Subject Editor)
openscience@royalsociety.org

Appendix A

In this paper, the team conduct a series of heroic field experiments attempting to measure the speed of individual blue and great tits to learn three spatial discriminations in which one of five automated feeders dispense a food reward. They collected tens of thousands of visits from hundreds of individuals. They then go on to explore what features of an individual or the experimental set up influence learning speed. This is where it gets messy, with various factors (sex, species, feeder position, feeder reliability) all having some effects in different tests. I have commented on a previous draft of this work and think that the methodology and analysis are fine. However, even with one aspect of the study (the social manipulation) moved to another MS, I still think that the current structure is sprawling and confusing and risks the great efforts involved collecting these data being wasted or at least not fully appreciated. Therefore, my comments are intended to offer suggestions to the authors as to how they may increase the focus of their MS so that others can benefit from it.

General comments: The Introduction does not draw the reader in. The authors are going on to ask series of important questions, but these are not clearly set out until L196-204. My reading of the Questions: 1) Are individuals repeatable in spatial discrimination tasks (do they exhibit consistent 'cognitive performance' or 'cognitive ability')? 2) What factors of the individual explain variation in learning speeds? 3) What factors of the environment, specifically the experimental setup, explain variation in learning speeds? For predominantly logistical reasons, the authors have focussed on 3 individual factors (sex, age, species) and X experimental set up factors (relative reward location, intertrial interval, mechanical reliability). There is an additional question that is somewhat tangential to these questions, namely what explains an individual's likelihood of participation in a test. I feel that the Introduction should follow the structure behind those questions (and consequently the rest of the MS) with: 1) a clear description of why it is important that any individual differences are consistent (although as I describe below, the three tests all draw on subtly different cognitive processes so perhaps we might not expect them to be very highly related), whether inconsistencies are the result of variations in ability or simply performance that is affected by immediate context (and an explanation early on of the difference between ability and performance, with consistent and appropriate use of the phrases throughout); 2) A description of what individual attributes might be expected to explain inter-individual diffs in CA or CP, with particular relevance to BT and GT, accompanied by clear, justified, predictions of what we might expect to see in this popn; 3) A description of what aspects of the experimental setup might be expected to explain inter-individual diffs in CA or CP, with particular relevance to BT and GT, accompanied by clear, justified, predictions of what we might expect to see in this popn. The work on social factors can be dropped from this MS and the participation results can be reported and discussed but are not central in my opinion. I also think that with so little variation in participation in Rev 1 & 2 (basically all birds that participate initially continue to do so) the authors could simplify their work by concentrating only on factors affecting participation in the first expt. Therefore, I think that the Introduction needs a substantial revision of structure.

I found the Results section very confusing. Given that consistency of performance is (rightly) deemed critical, I would expect to see that reported first. I suggest that one way to simplify the Results would be to have a Summary Table reporting what factors related to learning speeds in each test.

		Initial Discrimination	Rev 1	Rev 2
Blue Tit	Sex			
	Age			
	Feeder position			
	ITI			
	Feeder reliability			
Great Tit	Sex			
	Age			
	Feeder position			
	ITI			
	Feeder reliability			

This would also help the reader in the Discussion to see whether or not there are any general patterns.

The section on how position of feeder biases learning is very interesting, but seems more to do with mechanisms behind learning than an explanation of individual differences. I'm not sure that it currently adds much to the MS. I'd not weep if it were dropped from the MS and it could make for a nice standalone MS looking at how patterns of choice can give insights into cognitive processes.

Specific comments:

L58-60 Support with citation

L62 Are you talking about participation OR CP – previously it's been about participation.

L63 location..... of what? The test apparatus? The individual?

L64 Discrimination Expts MAY involve different spatial locations but may also involve discriminating other visual/aural/olfactory etc cues. Perhaps more generally introduce discrimination tasks, explaining that learning involves an increase in probability of selecting one option (location/cue) above a chance level. This can be assessed either by a criterion (e.g. X correct in a row) or a rate of improvement (a learning curve). THEN introduce the specific discrimination that you use based on cues of spatial location.

L67 Explain why edge feeders are easier to discriminate?

L70 Describe exactly what social information may be available. Seems to me that the most likely case in this expt is some form of local enhancement – more birds seen in an area makes it more attractive, rather than social learning as is usually construed in tits involving learning to imitate/emulate a particular action.

L85 This is a sudden introduction of the idea of differential learning strategies as indicated by visitation patterns – it's a nice idea and the data available mean that such questions could be addressed, but there has been nothing in the Intro so far to tell us about how individuals differ in such strategies, nor how they may relate to their performance. Drop or expand (another paper?)

L90-92 What is missing here are any predictions as to how an individual's CP might be expected to be determined by their age, sex, social status or species, specific to the tits being studied. E.g. do you predict that an adult female great tit should exhibit higher or lower CP than a juv male blue tit? Why?

L95 This is the first mention of reversals. Why are they being performed? How do they relate to the predictions about learning (that should be) made? I raised in a previous draft that they are not technically reversals where a cue's contingency is reversed – here at least some locations retain their affordances between trials. Briefly acknowledge that here but explain what aspects of a reversal task you are mimicking here (the need to suppress a previously learned association and replace it with a newly learned one). Perhaps move text from L186-190 to here. There needs to be an acknowledgement here or earlier that although reversals utilise at least some cognitive processes also used in discrimination learning, an individual's performance in such a task is also highly contingent on their executive function (inhibition) and perhaps their memory (forgetting). Performance in a second reversal may also be contingent on the ability to generalise ("Now I know that only one of these feeders will feed me...."). This introduction to reversal tasks and the idea that you are testing multiple cognitive abilities, even if they are not clearly defined) leads on to.....

L100 Quantifying individual consistency. Again, this needs an introduction as to why it is important, how it may be difficult to measure using cognitive tasks (see Cauchoix et al 2018), especially when comparing different tasks (discrimination vs reversal 1 vs reversal 2) which involve some shared and some different cognitive abilities.

L125 It is light after 16:00 in Jan/Feb. Did birds visit feeders after they were shut? In which case did this influence their performance in the same way as malfunctioning apparatus did as described later? It is possible that a visit to the location in expectation of a reward that was not rewarded may interfere with learning? Discuss.

L129 I think that the seed was visible but inaccessible behind the clear plastic door which had to be pushed open by the bird, but this only worked when the solenoid removed a lock? At the moment, it is unclear how the bird gets the reward and could sound like the solenoid drops a seed out to the bird. Clarify. Some further description in L144.

L131 'particle control board' – do you mean printed circuit board?

L141 Why the run for 7 days in one year and 8 in the other? It's not fatal, but please briefly explain.

L146 I agree no 'problem solving' required, but there could well have been a role for neophobia – unusual material transparent plastic etc?

L171-178 I'd open this section with the caveat that you didn't explore this treatment effect (it saves the reader concentrating too much on the detail, only to find it useless).

L196-204 This is a very helpful description of what was done and should really feature at the end of the Introduction – indeed these questions provide the basis for this paper and I suggest that the Introduction be restructured explicitly to lead up to these questions, with the supporting literature presented to justify each question.

L229-232 Given that by your definition a bird COULD learn the task in < 50 visits (one bird did so), what justifies your choice of 50 visits as the threshold for participation? Explain. Also need clarification that a bird could be a participant in one experiment, but not in another, although they

were only included as participants in later reversal experiments if they had participated in and learned the initial experiment (L234). This all gets a bit complicated.

L246 You've not yet described learning speed. Refer here to your description below.

L255-256 This is confusing – I think that you mean that whenever a bird visited an unrewarded feeder, it was counted as an error. You've already explained earlier that each bird was randomly assigned a rewarding feeder. Rephrase.

L263 I like the idea of testing the robustness of your criteria!

L282 Explain why you are including inter-trial interval as a factor here

L287-293 I think that fact that feeders malfunctioned and that this was an important predictor of 'learning speed' is a key finding of this work. I suggest that it is not buried here but given a separate heading.

L299 I don't understand why you excluded all correct visits? These are critical components of a bird's learning strategy and will highly likely shape their pattern of errors. E.g. if they had just made a correct choice, were they more likely to make an error by visiting a feeder immediately adjacent to their last visit, suggesting a small, rather than large spatial error? Perhaps explain here exactly what you can learn by only considering errors (and add what you could also learn by considering correct choices as well?)

Table 1 I think that some vertical lines to separate the 3 experiments might help here. Also in I356, emphasise that the nesting takes place across all experiments (an individual that drops out can never re-enter the analysis).

L368 'Almost all' - tell us how many

L369 I do worry that with only 14 non-participants and 204 participants in Reversal 1, these results are somewhat suspect and this gets 'worse' for Rev 2 with 4 non-participants vs 183 participants. What sort of effect sizes are you realistically likely to be able to detect with these skewed sample sizes? Is it worth dropping the analyses of Rev 1 & 2 and concentrating on the initial learning only?

L393 Quantify 'most individuals' – from the values given in I396-397 it actually looks like 67%-75% which is good, but 1/3-1/4 individuals 'revert' or never really 'learned it' initially.

L400 Here (and elsewhere) can you give some indication of the biological effect size as well as the statistical one – how much faster is -0.72 in relation to actual number of trials to reach criterion?

L399-431 I think that a Table here showing which Factors were sig predictors in each of the 3 experiments might help here (simple tick/cross one, with stats as currently in SI). It got rather confusing to keep checking back to see if one was or not in the previous experiment.

Fig 1 For clarity, are darker points representative of more data or is it just stylistic?

L458 'did not differ....' In what aspect?

L477 I thought that birds were biased AGAINST edge feeders in this expt, not towards them. Clarify

L529 – you conducted 2 reversal tasks, not 1

L552-556 I didn't follow the logic of this argument. Are you saying the all the non-participants were just in another flock from the participants? Non-residents? Clarify or cut.

L562 Emphasise this Is participation in the wild – lab studies usually better.

L571 This is critical – is there any chance that you could estimate the population ringed in the area that did not visit the feeders? (Not critical for this MS, but interesting),

L579 Describe what sort of cog task was used in the Brodin study – very different from your one. Especially given your (likely correct) conclusion that effects are task specific L583

L588-589 This difference between performance and ability should be clarified much earlier – see my general comments.

L606 I think that you need to be careful about describing ‘position in the flock’ based on the feeder location – surely it depends more on where the other birds are. Just say the position of the feeder relative to others differed.

L612-613 Explain here why a bias to edge feeders increases learning speed – I suspect that what it actually does is to bias their choice, making them more likely to visit the ‘correct’ feeder more often, and hence reach criterion faster, rather than actually learning faster (an non-cognitive explanation). This needs clarification and contrasting with the idea that they learn faster because the edge feeders are more conspicuous (a cognitive explanation).

L618 I’m not convinced that higher vigilance necessarily leads to lower cognitive performance – sure there may be lower attention, but equally there is an increased demand that the correct choice is made. The costs of getting it wrong may be higher, so learning is expected to occur more rapidly. Justify or remove.

L636 I’m not convinced that malfunction is simply overcome by larger samples. All this does is increase the risk that the CP of some (and perhaps in a large sample it becomes less clear WHICH) individuals will be confounded by technological glitches. For me, the take home message is that relying on automated data collection of CP risks producing very noisy and imprecise data. I think that this warning should be more prominent, especially as there appears to be a continuous drive towards greater automation in many aspects of behavioural study. It can be great, but is also high risk.

L640 I’d be cautious about describing CP across all 3 tasks as indicative of learning speed. Yes, that is one component, but as above, the Reversals also draw on executive functions and generalisations. Better, Blue tits showed consistency in their CP across all three experiments.

L647 Perhaps better to say, rather than ‘lower consistency’ that the performance of BT may be more susceptible to external factors such as the social environment and competition from GTs.

L649-651 This seems conjectural. I’d cut it.

L658-661 There seems to be a non-seq here. Indeed, these controls are hard to impose in the wild, but despite this, you find no effect of ITI. Your use of ‘and’ suggests that you would/should/did. Rephrase.

L677-678 I like this idea, and you can test this assumption. Presumably, you could check whether birds that had central rewarded feeders in the earlier tasks were more likely to be biased to central feeders later on?

L680 I don’t understand what you would gain by running tasks spanning ‘multiple heterogeneous environments.’ Clarify or cut.

L687 What do you mean that they 'remain relatively unaccounted for across the literature'? Most studies at least include some of these as Factors. Clarify.

L688 Here it is critical to differentiate between ability and performance – very different factors are likely to affect CA and CP.

Joah Madden 29 Jan 2020

Appendix B

We give our thanks to the reviewers and editor for taking the time to make constructive comments on our manuscript. Their effort is much appreciated, and the revision is certainly improved because of their work. Our responses to the individual suggestions are given in bold, below.

Associate Editor Comments to Author (Dr Alecia Carter):

Comments to the Author:

Dear authors, I have now received two constructive reviews of your manuscript, which you will find below. Both reviewers found the study to be well executed and interesting. I also found the study to be interesting, and the approach to be rigorous and thorough. In general, the suggested edits should improve the digestibility of the manuscript for readers.

I disagreed with the reviewers on one point, however: the priority and inclusion of the analysis of task participation. I agree with the authors that it is overlooked in studies of cognition and warrants investigation. It is not tangential—if there is low participation of a particular group of individuals in a cognitive task, the ‘remaining’ results on cognition will be biased. I also feel that, if an analysis is deemed necessary before the data come in, then it should be kept i.e. there should be no post-hoc changes in analyses to accommodate a ‘tidier’ story. That’s not necessarily the case here, as it is true that the sample of non-participants in the reversal-learning trials may be too small to detect an effect, as highlighted by both reviewers. But for the sake of analysis integrity, I would encourage the authors to be explicit in their revision about their initial analysis plans and their subsequent decisions.

(One question that remains is whether this measure of participation already misses some birds i.e. those tagged and alive but who don’t go to the feeders, as acknowledged in the discussion by the authors, albeit briefly. Can’t these data be analysed, post-hoc? Of course, the missing birds could be dead, but it still may have been useful information to include in this section of the MS, since the authors bring it up.)

Thank you for your editing and helpful comments. As described in more detail below, we continue to include participation as an important component of the manuscript. However, we agree with the reviewers that analyses of the factors influencing participation rates in the reversal learning experiments probably have insufficient power given the very small number of non-participants. So we have removed these, although the actual numbers of participants are still presented in the table. We address the question on how to account for birds that never visit the feeders in response to a similar comment by Reviewer 1, below.

Reviewer comments to Author:

Reviewer: 2

Comments to the Author(s)

I reviewed the manuscript “Determinants of individual variation in discrimination learning performance in wild mixed species flocks” for potential publication in Royal Society Open Science. The authors report some behavioral and demographic variables that affected performance on field-tested discrimination learning and reversal learning tasks in a mixed-species population. In my opinion, the presented data is scientifically sound and represent a valuable contribution to the field of behavioral ecology. Globally, I found the behavioral protocol and statistical analyses appropriate and in accordance with the standards in the field. I particularly liked the fact that they succeeded in testing a considerable sample size using a ‘classical’ discrimination learning task that normally requires a large time investment and which makes it rarely possible to test such high numbers.

Thank you, and we appreciate the time taken to read and make very helpful suggestions on the manuscript.

That being said, I do have some minor and some less minor concerns about a few aspects of the manuscript and analyses. I detail them in chronological order of the manuscript below.

Introduction

L51-62: I suggest to completely remove this section on the effect of participation on cognitive measurements. First, participation is expected to have a strong effect on the performance in problem-solving tasks, in which the number of interactions with the task inevitably increases chances of success. Most of the cited references (including [15], which is not a review) are in fact studies that examined, at least in part, problem-solving. For discrimination learning, I do not think that this is a “major source of bias” as suggested by the authors. It may have an effect, possibly through the mechanisms suggested at the end of the paragraph, but I do not think that it’s very likely and it’s somewhat speculative. Finally, as I will explain in detail below, I believe that there is insufficient data on participation in this study to draw any meaningful conclusion concerning its effect on learning performance. The participation rates (but not in models, see below) can still be presented in this manuscript as an information on the actual sample size, however, much less focus should be put on this aspect in the introduction, results and discussion.

We would disagree that participation is not a serious concern for discrimination learning experiments like ours. Because we are relying on voluntary visits by wild birds, we potentially have the same issue as the problem solving studies: if birds don’t visit the feeders, or don’t do so very often, they are not going to be able to learn the task. If certain types of individuals are less likely to visit than others, then this can introduce a bias into the dataset where we may make inferences about individual variation that are inappropriate because certain individuals were excluded or performed poorly because of differences in how they interact with the task rather than what they are capable of learning. So we continue to include this section, although we expand on our rationale to make it clear why participation is important.

However, we agree that the sample sizes are very low for participation in the reversal learning experiments, and that probably impacted our ability to come to meaningful conclusions for these analyses. There is no such concern for the initial learning experiment, where we had a robust sample size. Therefore, we have removed the analyses of the reversal learning experiments, aside from pointing out in the text and table that most birds continued to participate (and for that reason there was not enough variation to analyse the non-participants of these experiments).

The reference [15] (van Horik et al. 2017) is indeed an empirical paper, but within this paper they also review what is known about participation in various cognitive assays in the wild. Rather than repeat their findings and list of sources, we prefer to acknowledge their work in gathering this information and consider this the definitive review of the literature to back up our statement that participation is often low.

L80, and throughout the rest of the manuscript: I am puzzled as to why this experiment is not qualified as “spatial learning”? There are a few different ways to measure spatial learning, and the method described here appears to be one of them to my knowledge. If there is a good reason to avoid using the

term spatial, I would suggest mentioning it in the manuscript to help the reader understand why this choice was made.

We added a brief description for what we mean by discrimination learning (lines 75-78, 92-95, 99-102) and why we use this term (see also similar comment by Reviewer 2). Although we think the learning was likely based on spatial position of the feeder, this was also not necessarily the case, and so we use the broader and more conservative term discrimination learning.

L82: I suggest to remove “participation”.

See response to similar comment above.

L67: It would increase the credibility of this sentence if one or more references that investigated a more similar model/context were added.

We agree that this would improve the credibility of this conjecture, but we are not aware of any other studies that more directly relate to the context in which we are investigating (indeed, these types of studies are still rather rare in the literature and most systems have not advanced to the point where they are investigating and directly manipulating variables like predation risk in wild populations).

L90-92: The cited reference does not address the question of the effect on discrimination learning on survival and reproduction. There is in fact no mention of learning or any other cognitive trait in this article. I therefore suggest remove it. Unless other references are found to support it, this sentence should perhaps also be removed.

We agree that the cited reference was only indirectly relevant and have removed it. However, we think the sentence stands without a reference because it is a prediction rather than a statement of fact.

L99-100: Without further explanations, it is impossible to understand what this sentence refers to. I would suggest either removing it, or providing a little more information so that the reader has at least an idea of what this is about.

We removed this sentence.

L100-102: The “consistency” effect is problematic since it is measured using 3 different assays (even if they are measured using the same apparatus) that are normally assumed to measure different cognitive traits. Moreover, assuming that “consistent” measurements would suggest that those traits are targets of selection is very speculative. I suggest eliminating this analysis and interpretation. On the other hand, examining the relation between the three learning phases using appropriate models (e.g. linear regression) is indicated.

We agree that it was not quite correct to be talking about heritability/selection when we don't have repeatabilities in the sense of measuring exactly the same thing twice. We have removed these interpretations for the most part (although we do still discuss this in the more general sense in the introduction, just don't make any conclusions about heritability from our own results). However, the consistency of performance in these three assays is still of interest, and is suggestive of between individual differences, as we now detail in the Introduction.

Although this may not strictly be defined as “repeatability”, using the statistical method of calculating repeatability coefficients has value here because in essence we are measuring the same behavioural response (number of visits to meet a criterion) across different contexts. This is consistent with the use of this statistical approach in the behavioural ecology literature. So long as we are clear about what it means, we think this is valid heuristically. Note that since the two traits are subtly different, the phenotypic correlation implied by the individual variance component is suggestive of genetic covariation rather than heritability, though two measures of each trait would be needed to show the correlation is ‘repeatably’ or, perhaps, more consistent. Note also that simple correlations are less convincing because they don’t control for effects of fixed effects on both y and x variables, which the random term does, thus controlling for confounding effects that can lead to pseudorepeatability.

Methods

L146-149: If the data is available, I would be interesting to test the effect of the distance between the birds’ initial capture site and the experiment location on the learning performance.

We can see what this in principle would be interesting. However, we also think the analyses would most likely be problematic, or at a minimum overly complicated. First, the subjects were a heterogenous mix of birds that had last been captured as nestlings, and others that had been captured many times as free-flying juvenile or adult birds. All of those variables would need to be accounted for. Second, extensive mist netting was performed at the sites that were eventually used for the experiment in order to tag as many birds as possible prior to the experiment. Previous mist netting efforts in the woods were performed without reference to our specific study sites, so the distribution of distances will be difficult to work with. Birds are attracted with artificial feeders to mist netting locations, so the exact location of capture does not necessarily indicate something about the bird’s typical location. Third, the birds are not stationary during the winter, and instead roam throughout the woods in search of food. Therefore, we have not included such an analysis in the revision.

L229-231: How was the minimum 50 trials to participation chosen? Why not simply retain all the birds that reached the learning criterion? And why was the 1-bird exception retained?

We were interested in participation as a distinct behaviour from meeting the learning criterion. Some birds may have visited many, many times, but not learned. This is different from birds that never learned because they made very few visits. The value of 50 was chosen because only one bird with fewer visits met the learning criterion, while most birds with more than 50 visits (see Table 1) met the learning criterion. Therefore, 50 was a reasonable choice for a number of visits in which there was sufficient opportunity for the birds to have learned the task. We retained the one bird that met the learning criterion in fewer than 50 visits because it did indeed learn, and we saw no reason to throw it out for doing so in relatively few total visits.

L257: I recommend to remove “This assumption is likely valid”, it is not necessary here.

Removed as suggested.

L287-293: I strongly believe that all the animals that had dysfunctional apparatuses should be removed from the analyses. It is safe to assume that a non-consistent opening of the correct feeder will have a considerable effect on learning speed. The strength of this effect could vary depending on the time

when it happened (the trial that the bird had reached when this occurred, or any other variable that could have had an effect on learning when the feeder malfunctioned). I think that such a fundamental flaw in the behavioral protocol cannot be “corrected for” and therefore all those animals should be discarded.

We agree that inconsistent operating of feeders added noise to the data and may have affected individual’s experiences when interacting with devices in a way that could influence learning speed. And indeed we discussed earlier amongst ourselves in much detail whether we should exclude these birds. However, we agree with Reviewer 2 that this is a reality in technology heavy field studies, and that it is important to show these effects. Most sites had a feeder go down at some point, in which case all the birds at a site were potentially affected (whether or not it was ‘their’ assigned feeder). So it’s not really feasible to remove all these individuals, and we present the data as they are, and correct for this as best we can. Another way to look at it is that the patterns we observe could easily mimic natural disruptions in learning, e.g., due to the temporary residence of competitors or predators. If we threw out data for all confounding effects that could operate in a natural setting, there would be no data left!

L331-334: As pointed out above, I do not think that this analysis is appropriate. If different traits are being measured (as acknowledged by the authors), then there is no such thing as “consistency”. Furthermore, the performance across tasks should not be compared using a repeatability tool. It would be more appropriate to assess the relation between the performance in each assays using correlations and/or linear regressions.

See our response to the similar comment above.

Results

Table 1: I do not think that this table is necessary as a main table. It could be added in supplementary information, or alternatively combined with another, more relevant, table.

We do think this table helps the reader keep track of the sample sizes, which otherwise would be cumbersome to report in every figure and statistical result. And it gives a good overall picture of the demographics of the experiment. So we’d like to keep it in the main text, but agree with both reviewers that additional tables would also help, and we have adopted one similar to that suggested by reviewer 1.

L368-373: If almost all birds that learned in the initial learning phase also participated in reversal learning, then I am not sure that it is relevant to try to explain this “variation”. There is an almost inexistent variation here (n=14/409 birds that participated in RL but not IL) so I doubt that this statistical analysis is sound. I suggest to remove it.

We did remove as suggested, because the sample size of nonparticipants is very low, so there was only limited power to detect any effects.

L382-397: I do not think that all this section is necessary.

One of the aims of our paper is to describe the study system and justify our measures of learning, which we intend to continue to use in future experiments as well as further analyses of this dataset.

Our experience has been that learning criteria are hotly debated and justifications for a particular criterion are not always justified in manuscripts. We include this section to establish that we did explore alternative criteria and found them reasonably similar, helping to justify our eventual choice of criterion.

L404-407: As per above comment, I would remove this.

See above and response to reviewer 1 on our treatment of the birds that experienced outages.

L418: I suggest to replace “but, in the first...” by “in contrast, there was no effect of the species on the reversal learning speed”.

We reworded to more clearly emphasize the contrast.

L419-420: The result that reversal learning speed was not related to initial learning is extremely surprising. There is very often a strong correlation between those measurements. I believe that this unexpected result should be more discussed in the discussion section.

We agree this is interesting and should be highlighted more. The literature seems to be all over the place in terms of whether these performances should be positively or negatively correlated, or in fact uncorrelated as we found. We add some material to the discussion about this (line 701-713).

L424: As per the above comment, I would remove the birds that experienced outages, therefore this variable could be removed from models.

See above and response to reviewer 1 on our treatment of the birds that experienced outages.

L440: I would recommend rearranging this section (by bias type instead of by learning type?) and maybe adding subtitles to help the reader follow this section. The different types of biases are easily confused with each other. I had to go through this section numerous times to fully understand what the results were. Maybe a table summarizing all those results would help. In fact, a table might be more useful than the Figures 2 and 3.

This was a good suggestion to rearrange by bias type and we have done so, and think it is much easier to read now. We have also added a summary table as suggested.

L467-471: I would not denote the visits to previously learned feeder a “bias”. It’s the very logic of this experiment and it is expected that the birds will visit the previously learned feeder first (retention phase), before eventually figuring out that it is not rewarded anymore. I suggest to reword.

It is indeed the logic of the experiment, but it is a bias also in the sense that the individuals visited this feeder more than would be expected from a random distribution of feeders. Certainly we would have been surprised if there had not been a pattern with greater visits to the previously rewarded feeder, but it is nevertheless useful to confirm that this was the case, and we think the word bias is used appropriately here.

Figure 5: Is this result obtained by the “repeatability” analysis? If yes, then it is misleading as the graph represents a regression (and as stated in the legend).

We clarified the legend. A scatterplot with a regression line is a good way to illustrate the fact that there is consistency between two measurements. They are indeed not the same analyses, which we now state explicitly, but the repeatability analysis is best for statistical testing and a scatterplot is useful for illustrating the data. Repeatability analyses are also illustrated with scatterplots.

L534: Innovation and discrimination learning are two distinct traits, therefore I do not think that comparing the demographic factors that affect them is appropriate here. I suggest removing this sentence or rewording it.

We agree that this was not relevant and removed that portion of the sentence.

L586-587: Despite the absence of statistical difference between the species' performance on RL1 and RL2, there is a pronounced shift in the species performance from IL to RL2. Great tits are better in IL but blue tits are better in RL2. They actually have opposite slopes throughout the stages, blue tits have a negative slope and great tits have a positive slope. I think that this is very interesting and that it should be analyzed using appropriate statistical analyses (comparing slopes and/or the difference between RL2 and IL).

This is a good point that we hadn't explored the species effect as much as we could. We should note that the raw averages are somewhat misleading here because the data for reverse learning 2 in great tits are heavily skewed by one bird with a very large value for learning speed (this is why we used $\ln(\text{learning speed})$ for analyses and graphs), and there is also a very large value for one blue tit in the initial learning experiment. When looking at medians, the blue tits essentially have a flat line across the three experiments (see new Supplementary Figure 2), while great tits have a positive slope, as noted, but this is mainly due to their rather low values for the initial learning experiment, which then stabilize at a higher value in the other two experiments. So this is essentially the result we report, that great tits do better than blue tits in only one of the experiments, initial learning. However, we agree this is an interesting finding, and we expand a bit now in this paragraph: "This difference between experiments arose primarily because great tits learned quickly in the initial learning experiment but took longer to learn in the reversal experiments, whereas blue tits' learning speeds were generally similar across all three experiments (Supplementary Figure 2)." We also added a new figure in the supplement that illustrates with boxplots and individual data points the learning speeds for individuals of each species in each experiment.

L623-636: This section could be removed if the birds that experienced malfunctions were discarded.

See above, and response to reviewer 1 for our treatment of birds that experienced malfunctions.

L649-651: I suggest to remove this sentence as it is very speculative, and I think that it is not suitable to use the word "brightest" in this context.

Removed as suggested by both reviewers.

Reviewer 1 (copied over from PDF, some formatting lost in the process)

In this paper, the team conduct a series of heroic field experiments attempting to measure the speed of individual blue and great tits to learn three spatial discriminations in which one of five

automated feeders dispense a food reward. They collected tens of thousands of visits from hundreds of individuals. They then go on to explore what features of an individual or the experimental set up influence learning speed. This is where it gets messy, with various factors (sex, species, feeder position, feeder reliability) all having some effects in different tests. I have commented on a previous draft of this work and think that the methodology and analysis are fine. However, even with one aspect of the study (the social manipulation) moved to another MS, I still think that the current structure is sprawling and confusing and risks the great efforts involved collecting these data being wasted or at least not fully appreciated. Therefore, my comments are intended to offer suggestions to the authors as to how they may increase the focus of their MS so that others can benefit from it.

Thank you (again!) for reading through this and giving very helpful suggestions.

General comments: The Introduction does not draw the reader in. The authors are going on to ask series of important questions, but these are not clearly set out until L196-204. My reading of the Questions: 1) Are individuals repeatable in spatial discrimination tasks (do they exhibit consistent 'cognitive performance' or 'cognitive ability')? 2) What factors of the individual explain variation in learning speeds? 3) What factors of the environment, specifically the experimental setup, explain variation in learning speeds? For predominantly logistical reasons, the authors have focussed on 3 individual factors (sex, age, species) and X experimental set up factors (relative reward location, intertrial interval, mechanical reliability). There is an additional question that is somewhat tangential to these questions, namely what explains an individual's likelihood of participation in a test. I feel that the Introduction should follow the structure behind those questions (and consequently the rest of the MS) with: 1) a clear description of why it is important that any individual differences are consistent (although as I describe below, the three tests all draw on subtly different cognitive processes so perhaps we might not expect them to be very highly related), whether inconsistencies are the result of variations in ability or simply performance that is affected by immediate context (and an explanation early on of the difference between ability and performance, with consistent and appropriate use of the phrases throughout); 2) A description of what individual attributes might be expected to explain inter-individual diffs in CA or CP, with particular relevance to BT and GT, accompanied by clear, justified, predictions of what we might expect to see in this popn; 3) A description of what aspects of the experimental setup might be expected to explain inter-individual diffs in CA or CP, with particular relevance to BT and GT, accompanied by clear, justified, predictions of what we might expect to see in this popn. The work on social factors can be dropped from this MS and the participation results can be reported and discussed but are not central in my opinion. I also think that with so little variation in participation in Rev 1 & 2 (basically all birds that participate initially continue to do so) the authors could simplify their work by concentrating only on factors affecting participation in the first expt. Therefore, I think that the Introduction needs a substantial revision of structure.

We agree that the structure of the introduction needed some work. We have heavily revised this section, largely along the lines suggested by the reviewer. This includes more clearly delineating the different analyses, why they are important and how they fit into the manuscript as a whole. We make a better argument for why participation is indeed relevant within this framework (while at the same time cutting back on the emphasis on this variable, as suggested. We also removed the analyses of factors explaining variation in participation in the reversal experiments). Meanwhile, we reduce the material on social factors.

We give more detail on why individual or experimental factors might be expected to influence performance one way or the other. However, we note that in general the field is not advanced enough for specific predictions to be made, certainly not predictions that are so specific to the two study species. Plausible hypotheses can be made in both directions about most of the effects we examine. For instance: great tits (males, adults) learn faster than blue tits because they are more competitive and control access to feeders. Or, great tits (males, adults) learn slower than blue tits because they are more competitive and don't need to gain access to feeders because they can get food elsewhere. The effects are so complicated here that even with the controls included in our experimental design, there are still many possible explanations for the outcomes, so we prefer to be conservative and not make these kinds of predictions. Otherwise we end up with an enormously complex set of predictions making the paper even more intangible.

I found the Results section very confusing. Given that consistency of performance is (rightly) deemed critical, I would expect to see that reported first. I suggest that one way to simplify the Results would be to have a Summary Table reporting what factors related to learning speeds in each test. This would also help the reader in the Discussion to see whether or not there are any general patterns.

We included the summary table as suggested. However, we don't report consistency results first because we think they are easier to interpret after we have given the context of the learning speed results. We did, however, move this section above the section on errors/biases to keep the learning speed results closer together.

The section on how position of feeder biases learning is very interesting, but seems more to do with mechanisms behind learning than an explanation of individual differences. I'm not sure that it currently adds much to the MS. I'd not weep if it were dropped from the MS and it could make for a nice standalone MS looking at how patterns of choice can give insights into cognitive processes.

We make a better argument now for why these results are indeed important for understanding individual differences. In any case, we agree that these are very interesting results and would prefer to present them along with the rest of the material in this paper.

Specific comments:

L58-60 Support with citation

We rephrased this so that it reads as a prediction rather than a statement of something that is known to be the case. We are not aware of any studies that have directly tested this.

L62 Are you talking about participation OR CP – previously it's been about participation.

We reworded to make this about cognitive performance (participation is also affected by these factors, but since participation is a component of performance, it is clearer to leave out explicit mention of participation in this sentence).

L63 location..... of what? The test apparatus? The individual?

Clarification added.

L64 Discrimination Expts MAY involve different spatial locations but may also involve discriminating other visual/aural/olfactory etc cues. Perhaps more generally introduce discrimination tasks, explaining that learning involves an increase in probability of selecting one option (location/cue) above a chance level. This can be assessed either by a criterion (e.g. X correct in a row) or a rate of improvement (a learning curve). THEN introduce the specific discrimination that you use based on cues of spatial location.

We introduce more directly now the idea of a discrimination learning experiment and (as also brought up by the other reviewer) point out that in our case it may have been a spatial discrimination, but that's not necessarily the case either.

L67 Explain why edge feeders are easier to discriminate?

We added a brief explanation

L70 Describe exactly what social information may be available. Seems to me that the most likely case in this expt is some form of local enhancement – more birds seen in an area makes it more attractive, rather than social learning as is usually construed in tits involving learning to imitate/emulate a particular action.

We drastically reduced the material on social learning, as it was not the focus of this manuscript.

L85 This is a sudden introduction of the idea of differential learning strategies as indicated by visitation patterns – it's a nice idea and the data available mean that such questions could be addressed, but there has been nothing in the Intro so far to tell us about how individuals differ in such strategies, nor how they may relate to their performance. Drop or expand (another paper?)

We expanded on this and integrated it with the material on confounding factors on learning, so that it is more clear how this relates to the rest of the analyses in the paper (line 71-86).

L90-92 What is missing here are any predictions as to how an individual's CP might be expected to be determined by their age, sex, social status or species, specific to the tits being studied. E.g. do you predict that an adult female great tit should exhibit higher or lower CP than a juv male blue tit? Why?

See response to similar comment above. Predictions could be made, but in general they could be plausibly made in either direction, and any given result has many interpretations, so we do not think that including these predictions adds any value to the work, and would add too much confusion.

L95 This is the first mention of reversals. Why are they being performed? How do they relate to the predictions about learning (that should be) made? I raised in a previous draft that they are not technically reversals where a cue's contingency is reversed – here at least some locations retain their affordances between trials. Briefly acknowledge that here but explain what aspects of a reversal task you are mimicking here (the need to suppress a previously learned association and replace it with a newly learned one). Perhaps move text from L186-190 to here. There needs to be an acknowledgement here or earlier that although reversals utilise at least some cognitive processes also used in discrimination learning, an individual's performance in such a task is also highly

contingent on their executive function (inhibition) and perhaps their memory (forgetting). Performance in a second reversal may also be contingent on the ability to generalise (“Now I know that only one of these feeders will feed me....”). This introduction to reversal tasks and the idea that you are testing multiple cognitive abilities, even if they are not clearly defined) leads on to.....

We now introduce reversals much earlier in the introduction and acknowledge that they may involve partly non-overlapping cognitive abilities relative to initial learning. In fact, we argue that examining consistency in performance is important in part to get a handle on possible covariation (or not) between performance in different tasks and what that may mean for underlying mechanisms. We prefer to leave the material describing how our reversal task with 5 choices differs from a 2-choice task where it is because otherwise we would need to give a lot of details about the experimental design in the Introduction, which we think are better saved for the Methods section.

L100 Quantifying individual consistency. Again, this needs an introduction as to why it is important, how it may be difficult to measure using cognitive tasks (see Cauchoix et al 2018), especially when comparing different tasks (discrimination vs reversal 1 vs reversal 2) which involve some shared and some different cognitive abilities.

We spend more time on consistency and why it is important in the 2nd to last paragraph of the introduction. This introduces consistency as one of the major questions of the paper, and describes the different predictions that can be made for how individual performance could covary across experiments, and how these estimates may or may not be affected by confounding variables.

L125 It is light after 16:00 in Jan/Feb. Did birds visit feeders after they were shut? In which case did this influence their performance in the same way as malfunctioning apparatus did as described later? It is possible that a visit to the location in expectation of a reward that was not rewarded may interfere with learning? Discuss.

We had a typo here in fact and the feeders were left open until 16:30. However, the reviewer’s point still stands. We add details to this section about how the visits dropped off rather steeply at the beginning and end of the day. Nevertheless, there may have been a few visits before and after this time, and this could have affected their experience with the reward contingencies as we now point out.

L129 I think that the seed was visible but inaccessible behind the clear plastic door which had to be pushed open by the bird, but this only worked when the solenoid removed a lock? At the moment, it is unclear how the bird gets the reward and could sound like the solenoid drops a seed out to the bird. Clarify. Some further description in L144.

We added additional details here to complement the description in the next paragraph.

L131 ‘particle control board’ – do you mean printed circuit board?

Yes we did, thank you for catching this.

L141 Why the run for 7 days in one year and 8 in the other? It’s not fatal, but please briefly explain.
We added this.

L146 I agree no 'problem solving' required, but there could well have been a role for neophobia – unusual material transparent plastic etc?

We added some material a few sentences earlier addressing the possibility of neophobia and point out that the extended period of feeder availability prior to the experiment measuring learning speed in part served to allow birds to overcome any initial neophobia towards the devices.

L171-178 I'd open this section with the caveat that you didn't explore this treatment effect (it saves the reader concentrating too much on the detail, only to find it useless).

We moved the caveat up so that it is now more prominent near the beginning of this section.

L196-204 This is a very helpful description of what was done and should really feature at the end of the Introduction – indeed these questions provide the basis for this paper and I suggest that the Introduction be restructured explicitly to lead up to these questions, with the supporting literature presented to justify each question.

See response to similar suggestion above.

L229-232 Given that by your definition a bird COULD learn the task in < 50 visits (one bird did so), what justifies your choice of 50 visits as the threshold for participation? Explain. Also need clarification that a bird could be a participant in one experiment, but not in another, although they were only included as participants in later reversal experiments if they had participated in and learned the initial experiment (L234). This all gets a bit complicated.

We added this explanation: “We chose 50 as the cutoff for participation because only one of 21 birds that visited between 20-49 times met the initial learning criterion, while five of nine birds that visited between 50-79 times met the initial learning criterion.” There is no objective way to quantify when an animal is participating, but we think this is a reasonable approach. We also added a clarification about the nestedness of our data structure: that an individual had to be a participant and learn in a previous experiment before we included it in analyses of the subsequent experiment.

L246 You've not yet described learning speed. Refer here to your description below.

This ended up being removed since we no longer analyse factors affecting participation in reversal learning experiments.

L255-256 This is confusing – I think that you mean that whenever a bird visited an unrewarded feeder, it was counted as an error. You've already explained earlier that each bird was randomly assigned a rewarding feeder. Rephrase.

Rephrased to clarify as suggested.

L263 I like the idea of testing the robustness of your criteria!

L282 Explain why you are including inter-trial interval as a factor here

We added an explanation for this.

L287-293 I think that fact that feeders malfunctioned and that this was an important predictor of 'learning speed' is a key finding of this work. I suggest that it is not buried here but given a separate heading.

While we agree that it is an important finding, this was also an unplanned consequence of the experimental device, rather than an analysis that was integral to the original experimental goals. So we don't think it warrants a separate heading, which would almost make it look like this was an intentional experimental intervention.

L299 I don't understand why you excluded all correct visits? These are critical components of a bird's learning strategy and will highly likely shape their pattern of errors. E.g. if they had just made a correct choice, were they more likely to make an error by visiting a feeder immediately adjacent to their last visit, suggesting a small, rather than large spatial error? Perhaps explain here exactly what you can learn by only considering errors (and add what you could also learn by considering correct choices as well?)

We were looking at these errors as an aggregate across all of the bird's visits during an experiment. It could indeed be interesting to make a finer-scale analysis of when different kinds of errors occurred in relation to what the bird had done on its previous visit, or perhaps look for patterns in the sequence of errors across space and time. Such an analysis would be worth doing at some point, but is beyond the scope of this paper. Nevertheless, this is a reasonable question to ask, so we added a brief clarification of our goals for this analysis.

Table 1 I think that some vertical lines to separate the 3 experiments might help here. Also in l356, emphasise that the nesting takes place across all experiments (an individual that drops out can never re-enter the analysis).

We added the vertical lines, and point out now that the nesting is not just within but also across experiments.

L368 'Almost all' - tell us how many

We added these numbers to the text.

L369 I do worry that with only 14 non-participants and 204 participants in Reversal 1, these results are somewhat suspect and this gets 'worse' for Rev 2 with 4 non-participants vs 183 participants. What sort of effect sizes are you realistically likely to be able to detect with these skewed sample sizes? Is it worth dropping the analyses of Rev 1 & 2 and concentrating on the initial learning only?

We did just that, as suggested by both reviewers. We added an explanation for why we did not include similar analyses from the reversal experiments, because of this issue of small sample sizes of non-participants.

L393 Quantify 'most individuals' – from the values given in l396-397 it actually looks like 67%-75% which is good, but 1/3-1/4 individuals 'revert' or never really 'learned it' initially.

The values in (original) lines 396-397 are this quantification. We felt it less awkward to place these numbers at the end of the sentence rather than immediately after the phrase “most individuals”.

L400 Here (and elsewhere) can you give some indication of the biological effect size as well as the statistical one – how much faster is -0.72 in relation to actual number of trials to reach criterion?

We now include a supplemental figure showing raw data and boxplots for learning speeds for each species in each experiment. This is in addition to Supplementary Table 1, which shows average values of (raw) learning speeds for each of the demographic categories.

L399-431 I think that a Table here showing which Factors were sig predictors in each of the 3 experiments might help here (simple tick/cross one, with stats as currently in SI). It got rather confusing to keep checking back to see if one was or not in the previous experiment.

We agree this is helpful and have added such a table to summarise the results.

Fig 1 For clarity, are darker points representative of more data or is it just stylistic?

There is not a scale as such, but because the data points are partially transparent, if some of them continue to overlap despite the jitter, then they will end up darker. We added a brief statement in the figure caption to clarify this.

L458 ‘did not differ....’ In what aspect?

Rephrased to clarify

L477 I thought that birds were biased AGAINST edge feeders in this expt, not towards them. Clarify

Well spotted! Indeed, the bias favoured the centre feeder in this case.

L529 – you conducted 2 reversal tasks, not 1

Changed to clarify

L552-556 I didn’t follow the logic of this argument. Are you saying the all the non-participants were just in another flock from the participants? Non-residents? Clarify or cut.

We clarified this sentence (line 582-586).

L562 Emphasise this Is participation in the wild – lab studies usually better.

Added this emphasis, as suggested.

L571 This is critical – is there any chance that you could estimate the population ringed in the area that did not visit the feeders? (Not critical for this MS, but interesting),

We agree this would be very useful for our interpretation, and now include the following at the end of this paragraph: “To obtain a fuller picture of the factors influencing participation rates and their

consequences for estimates of variation in cognitive performance, it will be necessary to devise new methods to characterise movements and other behaviours of these non-participating individuals.” Because of the mobility of these birds and their fission-fusion flocks, we feel it would be too speculative to throw out numbers about how many may have been present, and would like to offer this up as an interesting potential follow up study that is badly needed (albeit very difficult to do well) in these types of studies in wild populations.

L579 Describe what sort of cog task was used in the Brodin study – very different from your one. Especially given your (likely correct) conclusion that effects are task specific L583

We now include a brief description, pointing out that the Brodin study was about observational learning of food cache locations.

L588-589 This difference between performance and ability should be clarified much earlier – see my general comments.

We now bring up cognitive performance and ability, and distinguish between them, very early on in the Introduction.

L606 I think that you need to be careful about describing ‘position in the flock’ based on the feeder location – surely it depends more on where the other birds are. Just say the position of the feeder relative to others differed.

This is a good point, and as we of course do not know where any non-visitors were at any given point, a feeder on the edge of the array may or may not have been at the edge of the flock. We rephrased to focus on the arrangement of feeders with respect to one another.

L612-613 Explain here why a bias to edge feeders increases learning speed – I suspect that what it actually does is to bias their choice, making them more likely to visit the ‘correct’ feeder more often, and hence reach criterion faster, rather than actually learning faster (an non-cognitive explanation). This needs clarification and contrasting with the idea that they learn faster because the edge feeders are more conspicuous (a cognitive explanation).

Yes, this was too subtly put in the original. We have revised to make a clearer distinction between the possibility that being assigned to a feeder on the edge makes for an easier learning challenge, and other possibilities (such as a general bias towards edge feeders).

L618 I’m not convinced that higher vigilance necessarily leads to lower cognitive performance – sure there may be lower attention, but equally there is an increased demand that the correct choice is made. The costs of getting it wrong may be higher, so learning is expected to occur more rapidly. Justify or remove.

We removed this because indeed the predictions aren’t clear, and as the reviewer pointed out earlier, the edge feeders may or may not represent the edge of the flock (which is what matters for predation).

L636 I’m not convinced that malfunction is simply overcome by larger samples. All this does in increase the risk that the CP of some (and perhaps in a large sample it becomes less clear WHICH)

individuals will be confounded by technological glitches. For me, the take home message is that relying on automated data collection of CP risks producing very noisy and imprecise data. I think that this warning should be more prominent, especially as there appears to be a continuous drive towards greater automation in many aspects of behavioural study. It can be great, but is also high risk.

We removed the sentence about sample sizes.

L640 I'd be cautious about describing CP across all 3 tasks as indicative of learning speed. Yes, that is one component, but as above, the Reversals also draw on executive functions and generalisations. Better, Blue tits showed consistency in their CP across all three experiments.

We rephrased this line and the next as suggested.

L647 Perhaps better to say, rather than 'lower consistency' that the performance of BT may be more susceptible to external factors such as the social environment and competition from GTs.

We rephrased as suggested to put the focus on factors that could explain variation in performance.

L649-651 This seems conjectural. I'd cut it.

Removed as suggested by both reviewers.

L658-661 There seems to be a non-seq here. Indeed, these controls are hard to impose in the wild, but despite this, you find no effect of ITI. Your use of 'and' suggests that you would/should/did. Rephrase.

Yes, good point, we rephrased to emphasize that we did not control for ITI, but that in any case there was no evidence that this variable had an effect.

L677-678 I like this idea, and you can test this assumption. Presumably, you could check whether birds that had central rewarded feeders in the earlier tasks were more likely to be biased to central feeders later on?

We added on some analyses as suggested. Indeed birds that had previously been assigned to the centre were more biased towards centre feeders in the next experiment (this is probably not surprising because we know that the birds were biased towards their previously rewarded feeder).

L680 I don't understand what you would gain by running tasks spanning 'multiple heterogenous environments.' Clarify or cut.

Removed, as suggested.

L687 What do you mean that they 'remain relatively unaccounted for across the literature'? Most studies at least include some of these as Factors. Clarify.

We removed this part and rephrased elsewhere to better represent the range of findings from previous studies.

L688 Here it is critical to differentiate between ability and performance – very different factors are likely to affect CA and CP.

We rephrased this sentence (line 719) to more clearly denote when we were talking about cognitive ability or cognitive performance, and have taken the reviewer's advice elsewhere to introduce this distinction earlier.

Joah Madden 29 Jan 2020